

# Measuring pH variability using an experimental sensor on an underwater glider

Michael P. Hemming[1], Jan Kaiser[1], Karen J. Heywood[1], Dorothee C.E. Bakker[1], Jacqueline Boutin[2], Kiminori Shitashima[3], Gareth Lee[1], Oliver Legge[1], and Reiner Onken[4]

[1]Centre for Ocean and Atmospheric Sciences, School of Environmental Sciences, University of East Anglia, Norwich Research Park, Norwich NR4 7TJ, United Kingdom.
[2] Laboratoire d'Océanographie et du Climat, 4, Place Jussieu, 75005 Paris, France.
[3]Tokyo University of Marine Science and Technology, 4-5-7 Konan, Minato, Tokyo 108-0075, Japan.
[4]Helmholtz-Zentrum Geesthacht, Max-Planck-Straße 1, 21502 Geesthacht, Germany.

*Correspondence to:* Michael P. Hemming (m.hemming@uea.ac.uk)

**Abstract.** Autonomous underwater gliders offer the capability of measuring oceanic parameters continuously at high resolution in both vertical and horizontal planes, with timescales that can extend to many months. An experimental ion sensitive field effect transistor (ISFET) sensor measuring pH on the total scale was attached to a glider during the REP14 - MED experiment in June 2014 in the northwestern Mediterranean Sea. During the deployment, pH was sampled at depths of up to 1000 m, along an
80 km transect over a period of 12 days. Water samples were collected from a nearby ship and analysed for dissolved inorganic carbon concentration and total alkalinity to derive pH for validating the ISFET measurements. The vertical resolution of the pH sensor was good (1 to 2 m), but stability was poor, and the sensor drifted in a non-monotonous fashion. In order to remove the sensor drift, a time-dependent, depth-invariant offset was applied throughout the water column for each dive, reducing the spread of the data by approximately two thirds. Furthermore, the ISFET sensor required temperature and pressure-based
corrections, which were achieved using linear regression. Correcting for this decreased the apparent sensor pH variability by a further 13 to 31 %. Sunlight caused an apparent sensor pH decrease of up to 0.1 in surface waters around local noon, highlighting the importance of shielding the sensor away from light in future deployments. The corrected pH from the ISFET sensor is presented along with potential temperature, salinity, potential density anomalies ($\sigma_\theta$), and dissolved oxygen concentrations ($c(O_2)$) measured by the glider, providing insights into physical and biogeochemical variability in this region. pH maxima
were identified at the depth of the summer chlorophyll maximum, where high $c(O_2)$ values were also found. Longitudinal pH variations at depth ($\sigma_\theta > 28.8\,\mathrm{kg\,m^{-3}}$) highlighted variability of water masses in this region. Higher pH was observed where salinity was $> 38.65$, and lower pH was found where salinity ranged between 38.3 and 38.65. It seemed that the higher pH was associated with saltier Levantine Intermediate Water. Furthermore, shoaling isopycnals closer to shore coinciding with low pH, high salinity, low $c(O_2)$ waters may be indicative of upwelling.





# 1   Introduction

It is estimated that a third of anthropogenic carbon dioxide emitted between 2004 to 2013 was absorbed by the oceans (Le Quéré et al., 2015). Normally unreactive in the atmosphere, carbon dioxide dissolved in seawater ($CO_2$(aq)) takes part in a number of chemical reactions. In particular, carbonic acid ($H_2CO_3$) forms as a result of ($CO_2$(aq)) reacting with water, which dissociates into bicarbonate ($HCO_3^-$) and carbonate ($CO_3^{2-}$). These seawater carbonate species are referred to as dissolved inorganic carbon concentrations $c$(DIC), with $HCO_3^-$ representing around 90 % of $c$(DIC). During the dissociation of carbonate species, hydrogen ions ($H^+$) are released. Since before the industrial revolution (year 1760), global surface ocean pH has fallen from 8.21 to 8.10 (corresponding to a 30 % increase in $H^+$ ion activity) as a result of the atmospheric $CO_2$ mole fraction increasing by more than $100\,\mu$mol mol$^{-1}$(Doney et al., 2009; Fabry et al., 2008). Future projections of anthropogenic carbon dioxide emissions suggest that ocean uptake of $CO_2$ will continue for many decades, thus contributing to long term ocean acidification (Rhein et al., 2013). This may have a significant effect on marine organisms, such as calcifying phytoplankton (e.g. coccolithophores) and corals (e.g. scleractinian), dependent on the solubility state of calcium carbonate (Doney et al., 2009).

Since 1989, it has been possible to measure pH to an accuracy of $10^{-3}$ using a spectrophotometric approach (Byrne and Breland, 1989). Although there have been some advances in adapting this method to measure pH autonomously *in situ* (Martz et al., 2003; Aßmann et al., 2011), it is largely used for shipboard measurements as it requires the use of indicator dye, and a means to measure spectrophotometric blanks, which is challenging outside of a laboratory (Martz et al., 2010).

pH is very dynamic, exhibiting diel, semi-diurnal, and stochastic variability at some locations (Hofmann et al., 2011), whilst varying on a seasonal (Provoost et al., 2010) to inter-decadal scale (Doney et al., 2009). A limited number of hydrographic surveys have been undertaken, and stations offering long term time series of pH are available (Rhein et al., 2013), but there is a drive to improve spatial and temporal data coverage via autonomous means, similar to what was experienced for temperature and salinity with Argo floats 16 years ago (Roemmich et al., 2003). There is demand to develop a reliable autonomous sensor with precision and accuracy of $10^{-3}$, whilst being affordable to the scientific community (Johnson et al., 2016).

Autonomous underwater gliders offer the possibility to observe the oceanic system with a greater level of detail on both temporal and spatial scales when compared with ship measurements (Eriksen et al., 2001). A low consumption of battery power and a great degree of manoeuvrability enable such vehicles to cover large areas and profile depths of up to 1000 m during missions that can last from weeks to months at a time. They are suitable platforms for a range of sensors, measuring both physical and biogeochemical parameters (Piterbarg et al., 2014; Queste et al., 2012).

The main goal of this paper is to describe the trial of a novel ion sensitive field effect transistor (ISFET) pH sensor which was attached to an autonomous underwater glider in the northwest Mediterranean Sea during the REP14 - MED sea experiment. The secondary objective is to provide a method of correcting pH measured by this sensor and to discuss the spatial and temporal variability observed. The experiment, ISFET sensor and the method of validation are described in Sect. 2. The initial pH results and validation, the method of further correcting pH, and an artifactual light-induced effect are described in Sects. 3.1 to 3.3. Corrected pH measurements are analysed alongside other collected parameters in Sect. 3.4, and finally, the paper's conclusions are provided in Sect. 4.



## 2 Methodology

### 2.1 REP14 - MED sea trial

This trial took place between 6[th] and 25[th] June 2014 in the northwest Mediterranean Sea off the coast of Sardinia, Italy (fig.1). This was part of the Environmental Knowledge and Operational Effectiveness (EKOE) research program led by the North Atlantic Treaty Organisation's (NATO) Centre for Maritime Research and Experimentation (CMRE), based in La Spezia, Italy. This was the 5[th] Recognised Environmental Picture (REP) trial, which was jointly conducted by two research vessels; the NRV *Alliance* and the RV *Planet*. More information about REP14 - MED is provided by Onken et al. (in preparation).

An iRobot Seaglider model 1KA (SN 537) with an ogive fairing operated by the University of East Anglia (UEA) was deployed within the REP14 - MED observational domain. The glider completed a total of 126 dives between 11[th] and 23[rd] June 2014. Out of this, the first 24 dives did not record pH and the last 9 dives were very shallow, leaving 93 usable dives. Successive dives were approximately 2 to 4 km apart, descending to depths of up to 1000 m. In total, twelve underwater gliders were deployed by participating institutions within the observational domain in an area of around 110 x 110 km$^2$.

### 2.2 ISFET and glider sensors

The ISFET pH sensor was stand-alone, meaning that the sensor was not integrated into any of the onboard glider electronics. It was situated next to a Seabird Conductivity-Temperature (CT) sensor (Fig. 2). The glider also carried a stand-alone ISFET $p(CO_2)$ sensor, and ISFET pH and $p(CO_2)$ sensors integrated into the glider's electronics (Fig. 2), but the retrieved data were of very poor quality and will therefore not be discussed in this paper.

To measure pH, the activity of hydrogen ions (commonly referred to as a concentration) is determined using the interface potential between the semiconducting ion sensing transistor coated with silicon dioxide ($SiO_2$) and silicon nitride ($Si_3N_4$), and the reference chlorine ion selective electrode (Cl-ISE). The ISFET pH sensor was previously found to have a response time of a few seconds and an accuracy of 0.005 pH (Shitashima et al., 2002; Shitashima, 2010). During the deployment, measurements were obtained every 1-2 m vertically. In addition to the ISFET sensors and the CT sensor, an Aanderaa 4330 oxygen optode was available to measure the concentration of dissolved oxygen ($c(O_2)$).

Measurements obtained by the ISFET pH sensor were converted from raw output counts to pH on a total scale using a two-point calibration with 2-aminopyridine (AMP) and 2-amino-2-hydroxymethil-1, 3-propanediol (TRIS) buffer solution carried out on deck at a surface air temperature of around 28 °C. These buffer solutions had a pH of 6.7866 and 8.0893, respectively. A linear fit using the raw output values measured from these buffer solutions before and after the deployment was used to convert the data series from the expedition to pH (Shitashima et al., 2002; Fukuba et al., 2008).

*In situ* temperature, practical salinity, and $c(O_2)$ measured by the glider have undergone a number of necessary corrections before analysis. An open-source MATLAB based toolbox (https://bitbucket.org/ bastienqueste/uea-seaglider-toolbox/) has been used to correct these variables (e.g. for differing timestamp allocations, sensor lags (Garau et al., 2011; Bittig et al., 2014), tuning the hydrodynamical flight model (Frajka-Williams et al., 2011)). Glider salinity and $c(O_2)$ measurements were also calibrated against inter-calibrated shipbourne CTD measurements obtained within the REP14 - MED observational domain.



### 2.3   Validation of ISFET pH measurements

As the *in situ* ISFET pH sensor was under trial, some form of validation of the results was required. In total, 124 water samples were collected from Niskin bottles sampled at 12 depths (down to 1000 m) using a CTD rosette platform at eight locations (casts 24 - 51) close to the glider's path (Fig.1). Water samples were collected between 05:19 Local Time (LT, UTC+2) on the

$9^{th}$ June and 16:58 LT on the $11^{th}$ June. The glider's ISFET pH sensor started operating at 16:36 LT on $11^{th}$ June. Overall, measurements obtained by the glider and the CTD overlap better in space than in time (Fig.3).

When collecting carbon samples, water was drawn into 250 mL borosilicate glass bottles from Niskin bottles on the CTD rosette using tygon tubing. Bottles were rinsed twice before filling and were overflowed for 20 seconds, allowing the bottle volume to be flushed twice. Each sample was poisoned with 50 $\mu$L of saturated mercuric chloride and then sealed using greased

stoppers, secured with elastic bands and stored in the dark (Dickson et al., 2007). The total alkalinity ($A_T$) and the $c$(DIC) of each water sample was measured in the laboratory using a Marianda Versatile INstrument for the Determination of Titration Alkalinity (VINDTA 3C - www.marianda.com). $c$(DIC) was measured by coulometry (Johnson et al., 1985) following standard operating procedure (SOP) 2 of Dickson et al. (2007), and $A_T$ was measured by potentiometric titration (Mintrop et al., 2000) following SOP 3b of Dickson et al. (2007). During the analytical process, 21 bottles of certified reference material (CRM)

supplied by the Scripps Institution of Oceanography, USA were run through the instrument to keep a track of stability and to calibrate the instrument. For each day in the lab, 1 CRM was used before and after the samples were processed. A total of 19 concurrent replicate depth water samples were collected, with around 2 to 3 replicates per CTD cast. On average, replicate water samples measured by the instrument differed by $(3.1 \pm 3.8)$ $\mu$mol kg$^{-1}$ for $c$(DIC) and by $(2.5 \pm 2.9)$ $\mu$mol kg$^{-1}$ for $A_T$, which equates to an error in pH of $(0.003 \pm 0.007)$.

Once $A_T$ and $c$(DIC) were known, pH could be derived using the CO2SYS program (Van Heuven et al., 2011). This calculation has an estimated probable error of around 0.006 pH due to uncertainty in the dissociation constants $pK_1$ and $pK_2$ (Millero, 1995). Temperature and salinity were obtained from the Seabird CTD sensor on the rosette sampler, and the seawater equilibrium constants were taken from Mehrbach et al. (1973) as refitted to the total pH scale by Lueker et al. (2000). More information on the equilibrium constants used in CO2SYS and other available carbonate system packages is described in Orr

et al. (2015). pH derived from water samples collected by ship and glider retrieved ISFET pH are both on the total pH scale (as described by Dickson (1984)), and will from now on be referred to as pH$_s$ and pH$_g$ respectively.

## 3   Results and corrections

### 3.1   pH validation

The objective of deriving pH$_s$ using $A_T$ and $c$(DIC) was to make a comparison with pH$_g$ measured by the ISFET sensor for

validation. Values of $c$(DIC) and $A_T$ were greatest at depths below 250 m, with lower values seen closer to the surface (Fig.4a-b), which is typical of the northwest Mediterranean Sea (Copin-Montégut and Bégovic, 2002). Mean $c$(DIC) and $A_T$ (averages over all casts) had standard deviations of 6.1 to 11.9 $\mu$mol kg$^{-1}$ and 5.9 to 10.6 $\mu$mol kg$^{-1}$, respectively, for the top 150 m of





the water column, and 1.7 to 3.9 $\mu$mol kg$^{-1}$ and 3.7 to 7.6 $\mu$mol kg$^{-1}$ for deeper waters, respectively. $pH_s$ had a maximum of 8.14 between 50 and 70 m depth (Fig.4c). Mean $pH_s$ had standard deviations of 0.004 to 0.011 within the top 150 m and 0.006 to 0.017 deeper than this. Only part of this variability is environmental; the remainder can be explained by the instrumental error discussed in Sect. 2.3.

Mean $pH_g$ and $pH_s$ agreed best between 70 and 250 m (Fig.4d), although a standard deviation of up to 0.03 existed at some depths in this range. Larger differences between these profiles can be seen above and below this depth range, with $pH_g$ 0.12 higher at the surface and roughly 0.1 lower between 950 and 1000 m when compared with $pH_s$. The $pH_s$ maximum at approximately 50 to 70 m depth was not apparent in the $pH_g$ profile, which was highest at the surface. The standard deviation values for $pH_g$ were large, with values of between 0.044 and 0.114 in the top 150 m of the water column and between 0.027 and

0.053 at other points in the water column. Comparing all $pH_g$ dive profiles obtained during the mission suggest a great degree of temporal and spatial variability, particularly at the surface, with pH ranging from 8.02 to 8.28. This range of pH is unlikely to be real as it is almost three times greater than the typical peak-to-peak amplitude of the annual pH seasonal signal found at the surface, and roughly five times greater than typical inter-annual pH variations found at greater depths in the northwest Mediterranean Sea (Copin-Montégut and Bégovic, 2002).

It was expected that the measurements from the glider matched those from the CTD, as any discrepancies between data sets would indicate possible instrumental or methodological issues. Mean profiles of potential temperature and salinity collected by the glider and by CTD (Fig.4e-f) agreed well, with values obtained by both methods being mostly within one standard deviation of one another. Mean values of potential temperature and salinity retrieved during CTD casts 24 to 51 ($pH_s$ casts) differed from the mean calculated using all casts at depths between 100 and 500 m. However, this is likely related to temporal

or spatial variability as values are within the range of all available glider measurements (grey area).

## 3.2   Effect of solar irradiance on the sensor

An apparent diel cycle in $pH_g$ anomalies (calculated by subtracting the all time mean from the hourly means within a given depth interval) was found predominantly at depths shallower than 20 m (Fig.5b). Lower pH obtained during both the ascending and descending parts of the dives was found between 09:00 and 18:00 local time (LT), decreasing by $> 0.1$ between 12:00

and 14:00 LT. Contrastingly, potential temperature, salinity, and $c(O_2)$ did not have strong diel cycles (Fig.5c-e), suggesting that this decrease in pH was not caused by changing environmental conditions. Particularly, one might expect $c(O_2)$ to have a similar pattern to pH if it was related to photosynthesis / respiration due to variations in $p(CO_2)$ (Cornwall et al., 2013; Copin-Montégut and Bégovic, 2002). However, $c(O_2)$ remained relatively constant throughout the day at all depth ranges implying that the level of biological activity in this region did not change on average throughout the day and hence would not have

caused this reduction in $pH_g$.

The decrease in $pH_g$ coincided with increased levels of solar irradiance (Fig.5a) recorded at meteorological buoy M1 (Fig. 1) during the day at the surface, hence it was likely a light-induced instrumental artefact. The effect of light on the voltage output of FET based sensors using $SiO_2$ and $Si_3N_4$ sensitive layers is known (Wlodarski et al., 1986), as the presence of photons can excite electrons in the valence band of the semiconductor material, creating holes and allowing the flow of electrons to





the conduction band. This increases the voltage threshold, falsely measuring higher hydrogen ion activity, leading to lower pH values (Liao et al., 1999).

This effect of light on our sensor was investigated further by exposing two ISFET pH sensors to artificial light whilst placed in reference buffer solutions (TRIS and AMP) in laboratory conditions. The results (not shown here) confirmed that our ISFET

sensor is affected by light when using either an LED or a halogen light source.

There were not enough dives for a robust light correction, and an irradiance measuring sensor was not attached to the glider, hence data collected within the top 50 m between 05:00 and 21:00 LT were not used in later analysis. In order to reduce this light effect on pH measurements in future, ISFET sensors will have to be placed on the underside of the glider or equipped with a light shield.

## 3.3  Correcting pH for drift, temperature and pressure

Overall, the range of $pH_g$ and its temporal and spatial variability were much greater than those of $pH_s$ at all sampled depths in the water column. Comparing mean profiles of $pH_g$ and $pH_s$ with one another indicated greater differences in the top 100 m where the temperature gradient of the water was strongest, and between 100 and 1000 m depth, where the difference changed more gradually as a function of pressure (Fig.4d). Therefore, it seemed that the ISFET pH measurements were not only related

to the amount of hydrogen ion activity in the water, but also to the temperature and pressure the sensor experienced.

The response of the ISFET sensor can be described by the Nernst equation (Eq. (1)), which relates sensor voltage to hydrogen ion activity:

$$E = E^* - S \lg( a(\mathrm{H}^+)\, a(\mathrm{Cl}^-)) \tag{1}$$

which incorporates the Nernst slope (Eq. (2)):

$$S = RT \ln(10)/F \tag{2}$$

where $T$ is temperature (k), $R$ is the gas constant ($8.3145\,\mathrm{J\,K^{-1}\,mol^{-1}}$), $F$ is the Faraday constant ($96485\,\mathrm{C\,mol^{-1}}$), $a(\mathrm{H}^+)$ and $a(\mathrm{Cl}^-)$ are the proton and chloride ion activities, $E$ is the measured voltage by the sensor (i.e. electromotive force), and $E^*$ is representative of the two half-cells in the ISFET sensor forming a circuit (i.e. interface potential) (Martz et al., 2010). It is known that temperature and pressure have an effect on $E^*$ (strong linear relationship), and that the Nernst slope is a function

of temperature. Also studies have shown that it is possible for ISFET sensors to experience some form of hysteresis as a result of changing $T$ and $P$ (Martz et al., 2010; Bresnahan et al., 2014).

The first step of correction involved reducing the unrealistic scale of $pH_g$ to something closer to the range of $pH_s$ eliminating in part the effect of instrumental drift. A constant-depth time-varying offset correction was applied using the difference between mean $pH_s$ and each $pH_g$ dive measurement where potential temperature was 14.0 °C, as water with this temperature

was situated at a depth below the thermocline where the potential density gradient was weak. This constant offset correction





decreased the range of pH measured by the ISFET sensor by approximately two thirds (Fig.6), with new $pH_g$ standard deviations ranging between 0.009 and 0.048 shallower than 150 m, and between 0.008 and 0.017 throughout the rest of the water column.

After applying this constant offset correction, a selection of dives was used to determine coefficients for further correcting $pH_g$ for potential temperature and pressure. 16 dives were selected for the following reasons: (1) they had similar temperature gradients in the upper part of the water column, where temperature was expected to have the greatest effect on measurements, (2) their locations were spread out along the glider's transect path, limiting area specific bias (i.e. any differences between measurements obtained in the open ocean and closer to the shelf), and (3) they contained measurements down to more than 900 m depth for a full-range pressure correction. Linear regression models were used to identify relationships between $pH_g$ and *in situ* temperature, and pressure. A brief outline of the method used is described below:

1. Calculate $\Delta pH$ as the difference between mean $pH_s$ and $pH_g$, using sample points close to one another in depth space.

2. Determine the line of best fit between $\Delta pH$ and *in situ* temperature in the top 100 m of the water column where the temperature gradient was strongest using linear regression.

3. Correct $pH_g$ for *in situ* temperature for the entire water column using the slope ($m$) and intercept ($c$) coefficients of the best fit line in step 2. to obtain $pH_{g,Tc}$, where '$Tc$' stands for 'Temperature corrected' values.

4. Calculate the difference between $pH_{g,Tc}$ profiles and mean $pH_s$, producing $\Delta pH_{Tc}$.

5. Determine the line of best fit between $\Delta pH_{Tc}$ and pressure for the lower 900 m of the water column using linear regression.

6. Correct $pH_{g,Tc}$ for pressure for the entire water column using coefficients $m$ and $c$ in a similar way to step 3. to get $pH_{g,TPc}$, where '$TPc$' stands for 'Temperature and Pressure corrected' values.

Good fits were found between $pH_g$ and potential temperature, and pressure (Fig.7). The standard deviations of $pH_{g,TPc}$ ranged between 0.008 and 0.039 in the top 150 m of the water column and between 0.007 and 0.013 at greater depth, a further decrease in apparent variability of 13 to 23 % and 14 to 31 % respectively (Fig.6).

Salinity covaries with temperature and pressure, and some of the salinity dependence of the offset between $pH_s$ and $pH_g$ might have been mis-attributed to the regression coefficients associated with temperature and pressure. The sensor characteristics should therefore be studied in detail under controlled laboratory conditions. However, for the purposes of calibrating the high-resolution, but poor-accuracy measurements obtained from the ISFET pH sensor, the present empirical correction based on temperature and pressure appears to be sufficient to achieve a match to within the pH repeatability of the discrete samples of between 0.004 and 0.017.



## 3.4 pH variability

Spatial and temporal variability can be seen in $pH_{g,TPc}$ (fig.8a-f). In the top 100 m, pH higher than 8.12 was found at depths ranging from 20 to 95 m, whereas lower values ranging from 8.06 to 8.09 were present closer to the surface at some locations. pH maxima were found at depths between 40 and 70 m, where potential temperature was around 15 °C (Fig.9a-c), within the

pycnocline (Fig.9g-i). A Deep Chlorophyll Maximum (DCM) is common at these depths in the Mediterranean Sea when waters are thermally stratified (Estrada, 1996). There is a build-up of actively growing biomass with greater cell pigment content as a result of photoacclimation, due to there being increased concentrations of nitrate, phosphate, and silicate, as well as sufficient levels of light at these depths (Estrada, 1996). This band of high pH situated at 20 to 95 m depth corresponded with a thick layer of $c(O_2)$ rich water at the same depths (Fig.9j-l). This high $c(O_2)$ is likely related to photosynthesis, particularly at the DCM,

where $p(CO_2)$ would be used up increasing levels of pH (Cornwall et al., 2013; Copin-Montégut and Bégovic, 2002). Below this band, $c(O_2)$ decreased to a minimum of $< 170\,\mu mol\,kg^{-1}$, which corresponded with lower pH. This low pH, low $c(O_2)$ water was likely a result of increased respiration and remineralisation at depth (Lefèvre and Merlivat, 2012). Furthermore, as described in Sect. 3.1, $c(DIC)$ concentrations, which have a strong relationship with $p(CO_2)$ (Merlivat et al., 2015), were higher below 200 m depth which would support this notion.

The transects can be separated into two regions roughly either side of 7.7° E for depths greater than 100 m. The western part containing lower pH of between 8.05 and 8.1, and the eastern part where higher pH ranging from 8.07 to 8.11 was found. The spatial variability of these two regions differed for each time period, with both the eastern high and western low pH patches changing in size vertically and horizontally. Salinity, potential temperature, and $c(O_2)$ were lower in the western part at a range of depths, compared with values found at similar depths in the eastern section (Fig.9a-f, 9j-l). The difference in pH between the

eastern and western parts highlights the variability of water masses in this region. In particular, the higher pH found at depth in the eastern part of the transect was likely related to the flow of Levantine Intermediate Water (LIW), which is typically found at depths of between 200 and 800 m close to the shelf slope (Millot, 1999).

The pycnocline shallowed between 7.7 and 8.16° E in the top 100 m of the water column during all three time periods, which corresponded with shoaling high salinity, lower pH, low $c(O_2)$ waters. These features may be related to upwelling.

Meteorological buoy M1 located south of the glider's transect recorded an average surface wind direction of 198° towards the south-southwest which would be favourable for coastal upwelling, however, the mean wind speed was only 2 m s$^{-1}$ which is weak. On the other hand, salinity maxima seen at depths of 200 to 700 m seem to suggest a spreading of the LIW westward. Such encroachment has been shown to increase divergence in regions close to shore with strong alongshore currents (Roughan and Middleton, 2004; Roughan et al., 2005). Upwelling signatures at this longitudinal range along the Sardinian coast have

been simulated, particularly in the summer, by Olita et al. (2013) using a hydrodynamic 3D mesoscale resolving numerical model. They suggest a mixture of both current flow and wind preconditioned and enhanced upwelling in this region, which may have also been the case during our deployment.



## 4  Conclusions

Our trials of an experimental pH sensor in the Mediterranean Sea uncovered instrumental problems that will need to be addressed in future usage. These are summarised here:

1. The sensor was subject to drift. This could be reduced by subtracting a time-varying constant-depth offset from each dive using the difference between $pH_g$ and $pH_s$ at a more dynamically stable depth, but such an approach is not generally recommended or valid.

2. The sensor was apparently affected by temperature and pressure, but it is unclear to what extent the empirical relationship between temperature and pH bias in the thermocline (top 100 m) and between pressure and pH bias in the deeper water (100 - 900 m) can be generalised.

3. The effect of light caused the sensor to measure lower levels of $pH_g$ in surface waters. This effect is expected to be ubiquitous wherever the sensor nears the surface during daytime. In future, the sensor will have to be positioned on the underside of the glider or equipped with a light shield to limit the effect of the sun when close to the surface.

Despite the overall disappointing performance, we were able to demonstrate the use of the corrected glider pH measurements for uncovering biogeochemical variability associated with biological and physical mesoscale features.

*Acknowledgements.* The authors would like to thank all partners who helped make REP14 - MED a success, the engineers, technicians and scientists onboard the NRV *Alliance*, NRV *Planet*, and those on land responsible for the logistics of the experiment, and the UEA glider science team for piloting the glider. We thank Bastien Queste and Gillian Damerell for help and support regarding the analysis of glider data. Michael Hemming's PhD project is funded by the Defence Science and Technology Laboratory (DSTL, UK) in close co-operation with Direction générale de l'armement (DGA, France), with oversight provided by Tim Clarke and Carole Nahum. We thank the Natural Environment Research Council (NERC, UK) for providing financial support for the demonstration of glider capability.



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



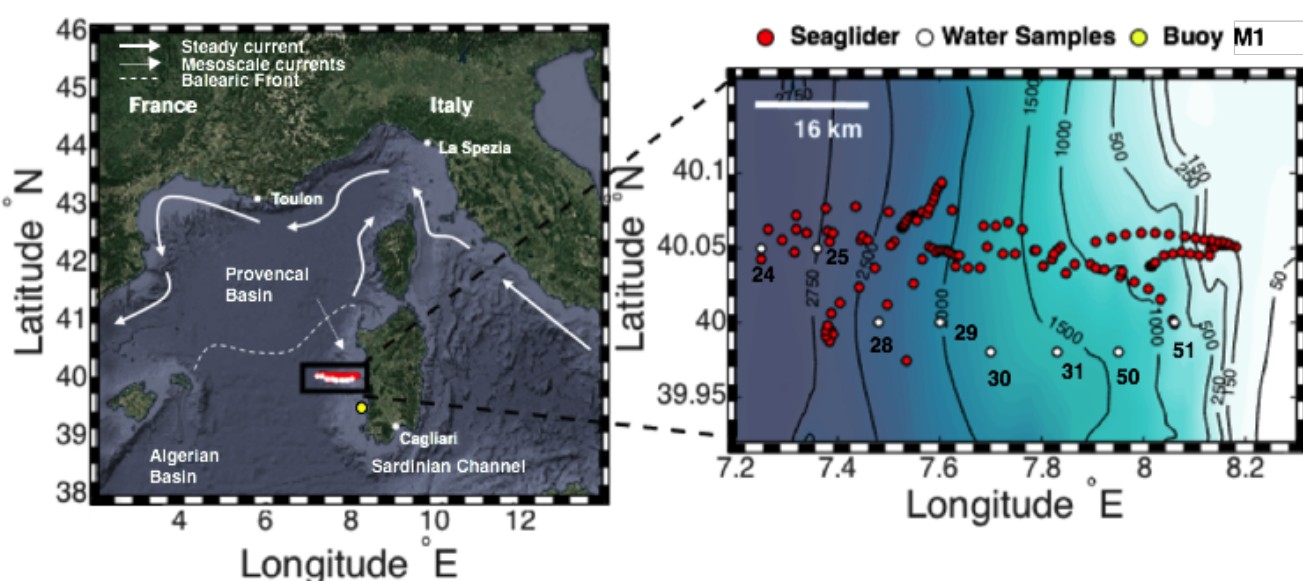

**Figure 1.** The locations of the 93 dives undertaken by the Seaglider (red markers), the 8 numbered ship CTD casts in which water samples were obtained (white markers), and Meteorological buoy M1 (yellow marker) within the REP14-MED observational domain off the coast of Sardinia, Italy between 11$^{th}$ and 23$^{rd}$ June, 2014. GEBCO 1 minute resolution bathymetry data (metres) were used (http://www.bodc.ac.uk/projects/international/gebco/), and surface circulation patterns were adapted from Millot (1999).





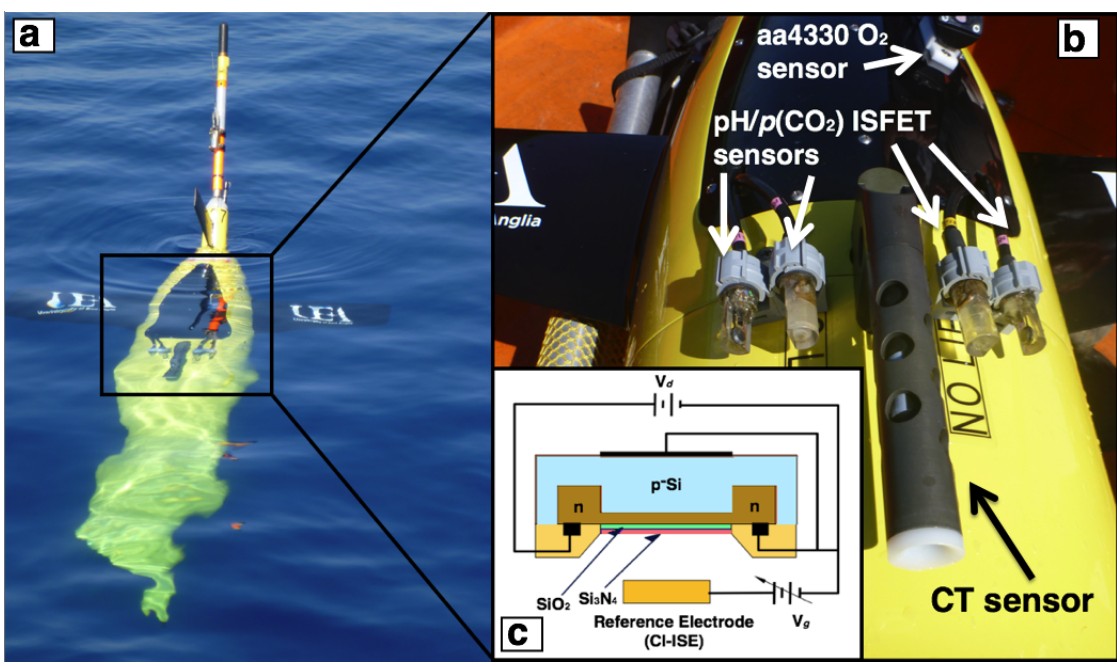

**Figure 2.** (**a**) Seaglider SN 537 during deployment, (**b**) a close up of the sensors, and (**c**) a schematic diagram of the ISFET sensor adapted from Shitashima (2010).





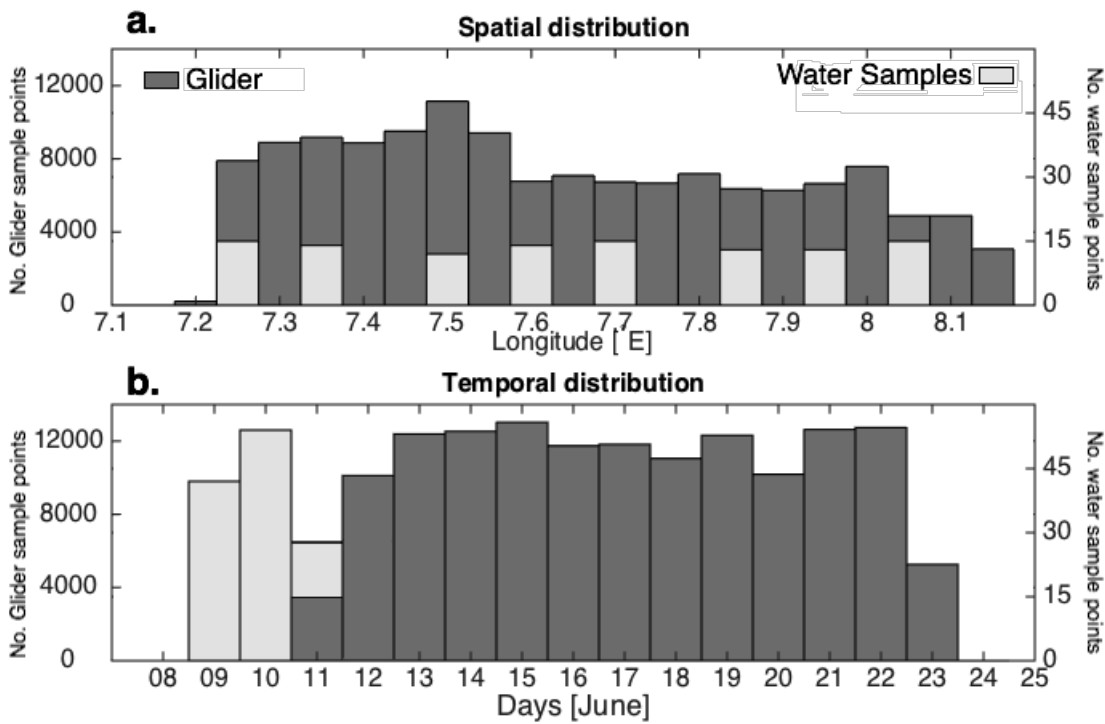

**Figure 3.** Histograms showing the (**a**) spatial and (**b**) temporal distribution of samples collected by the glider (dark grey) and by CTD water bottle sampling (light grey). The y-axis on the left is for the sum of glider samples, whilst the y-axis on the right is for the sum of water samples.



**Figure 4.** (**a**) Measured dissolved inorganic carbon concentrations $c$(DIC), (**b**) total alkalinity ($A_T$), and (**c**) derived total pH (pH$_s$) for each of the eight CTD cast locations (24 - 51). (**d**) Glider total pH (pH$_g$) profiles (grey markers) are displayed in the background, with the depth binned mean profile of these dive profiles displayed in the foreground (red markers) alongside the pH$_s$ depth binned mean profile (white markers). Standard deviation values are displayed as coloured error bars. (**e**) Potential temperature ($\theta$), and (**f**) salinity ($S$) profiles retrieved from the glider (grey markers) are displayed in the background, whilst their depth binned mean profiles and standard deviation error bars are superimposed on top. A comparison is made between the depth binned mean profile of the CTD measurements obtained during the eight casts (white markers, 7.2° - 8.1°E, 39.7° - 40.05°N), all of the measurements obtained from within the REP14 - MED observational domain (yellow markers, 7.2° - 8.4°E, 39.2° - 40.2°N, area of 110 x 110 km$^2$), and all of the glider's data points (red markers, 7.2° - 8.2°E, 39.95° - 40.1°N).




**Figure 5. (a)** Solar irradiance measured using a pyranometer on Meteorological buoy M1 (Fig.1), **(b)** glider retrieved pH (pH$_g$), **(c)** dissolved oxygen concentrations ($c$(O$_2$)), **(d)** potential temperature ($\theta$), and **(e)** salinity ($S$) average anomalies (calculated by subtracting the all time mean from the hourly means within a given depth interval) for each hour of the day local time (LT) for five near-surface depth ranges; < 5 m, 5 - 10 m, 10 - 15 m, 15 - 20 m, and 20 - 50 m, and 2 deeper depth ranges; 50 - 100 m, and 100 - 1000 m, for both ascending (upward triangle) and descending (downward triangle) dive profiles. The grey shaded area represents the nighttime, whilst the lightly shaded area represents the daytime.



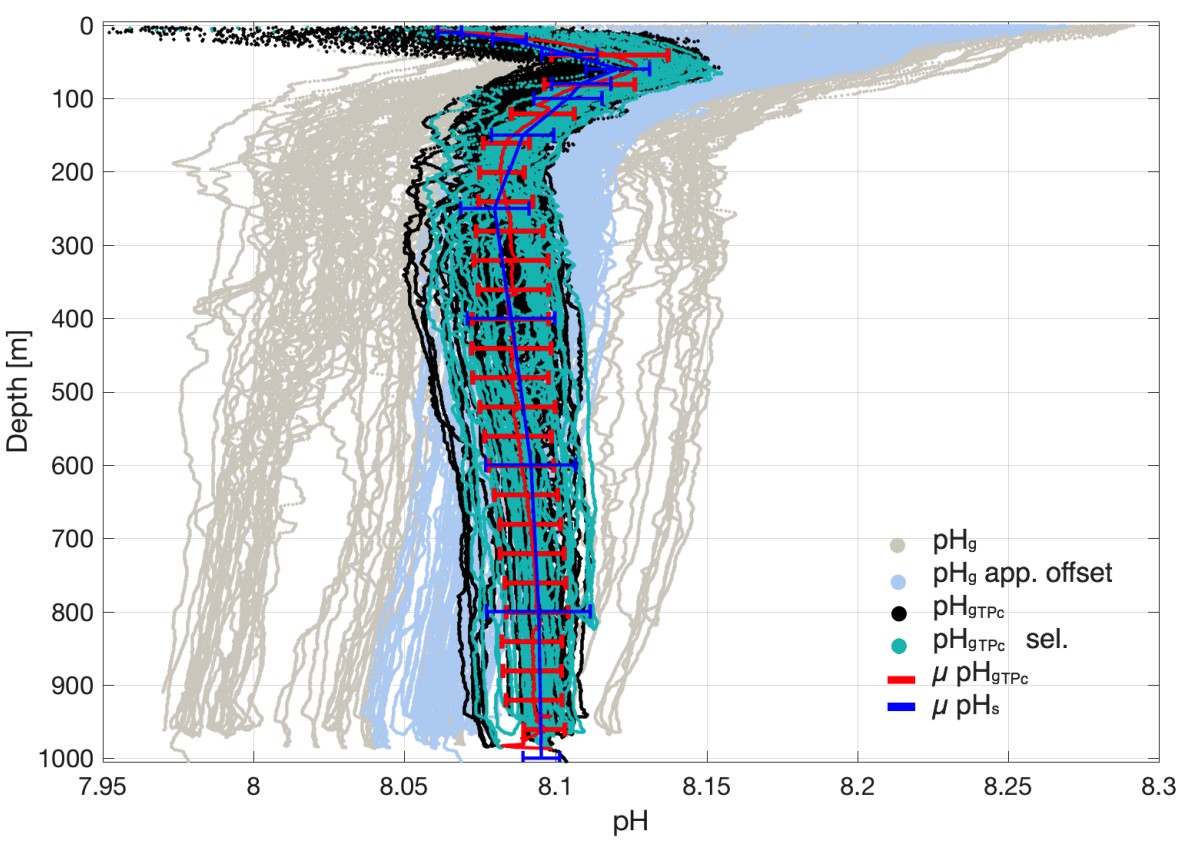

**Figure 6.** Profiles of $pH_g$ pre-correction (light grey), with an offset correction applied (light blue), with *in situ* temperature and pressure corrections for all dives (black), and for the 16 dives selected for the correction process (turquoise), are displayed. The depth binned mean profile of drift, temperature and pressure corrected $pH_g$ is shown in the foreground (red) along with the standard deviation ranges every 50 m. The depth binned mean $pH_s$ profile is plotted for comparison (blue) with standard deviation ranges at each sampled depth.





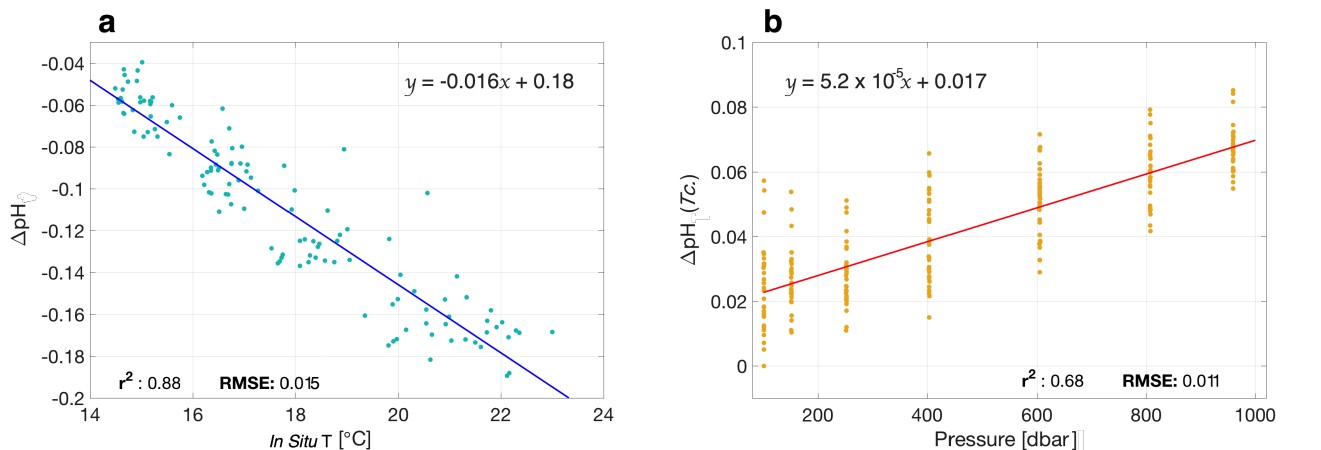

**Figure 7.** Linear regression fits are displayed for (**a**) $\Delta$pH (difference between $pH_s$ and $pH_g$ corrected for drift) vs. *In Situ* temperature in the top 100 m of the water column, and (**b**) $\Delta$pH corrected for *In Situ* temperature ($\Delta pH_{Tc}$) vs. pressure between 100 and 1000 m using selected dives. The $r^2$, the root mean square error (RMSE), and the equation of the line are displayed for each linear fit.





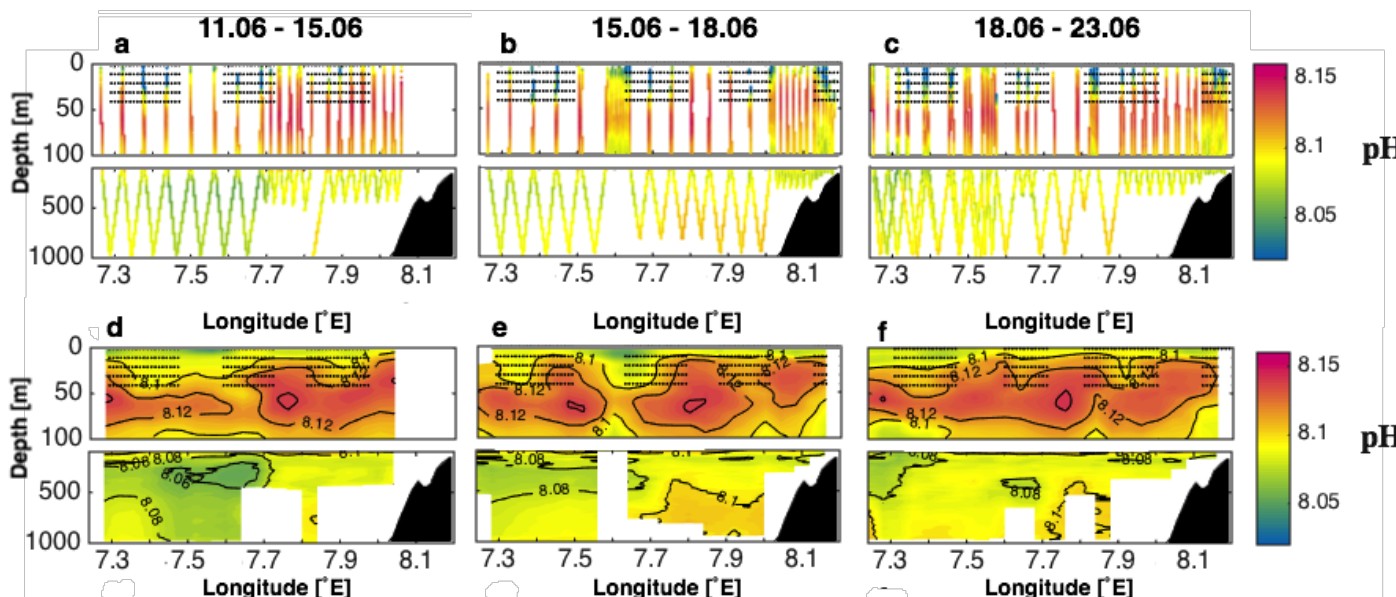

**Figure 8.** (**a-c**) Scattered $pH_{g,TPc}$ data used for objective mapping, and (**d-f**) the corresponding transects of objectively mapped $pH_{g,TPc}$, displayed spatially for three different time periods, with each period roughly corresponding to a completed transect. Data affected by light in the top 50 m and not used in optimal interpolation are overlaid by black markers. Bathymetry is coloured black, using GEBCO 1 minute resolution bathymetry data. Glider data were gridded into 2 m x 0.05° Longitude bins, and the radius of influence used for objective mapping was 0.1° longitude, 10 m vertically. 'Classical' Gaussian weighting was used during the mapping process.





**Figure 9.** Objectively mapped transects showing (**a-c**) potential temperature ($\theta$), (**d-f**) salinity ($S$), (**g-i**) potential density anomaly($\sigma\theta$), and (**j-l**) dissolved oxygen concentrations ($c(O_2)$), against longitude for three different time periods, with each period roughly corresponding to a completed transect. The spatial distribution of the data used for mapping these variables are exactly the same as that shown for pH in Fig.8a-c. Bathymetry is coloured black using GEBCO 1 minute resolution data, and the glider data used for mapping were binned into 2 m x 0.05° longitude grids. The radius of influence used the same dimensions as these bins and incorporated a 'Classical' Gaussian weighting distribution.