# Peer review of "Measuring pH variability using an experimental sensor on an underwater glider"

_Ocean Science, 2016_

## Referee Comment (RC1) · Anonymous Referee #1 · 15 Nov 2016

Published: 11 October 2016

Review of Hemming et al.: Measuring pH variability using an experimental sensor on an underwater glider

Special issue: REP14-MED: A Glider Fleet Experiment in a Limited Marine Area

GENERAL COMMENTS

This work is a contribution to a special issue about an observational experiment in a limited area in the western coast of Sardinia, where two research vessels collected high resolution oceanographic data using different observational systems as described in Onken et al. (same special issue). A fleet of 11 gliders was deployed. The current manuscript by Hemming et al. (OSD, 2016) deals with the results from one of them equipped with a pH sensor. The relevance of the manuscripts derives from the clear need by the biogeochemical community devoted to study the carbon cycle to implement reliable biogeochemical sensors in autonomous platforms. This sensors, pH included, should be developed and tested in the lab, and finally in a variety of ocean conditions, from coast to open ocean, surface to deep, high and low saline, etc..

To be concise I consider Hemming et al. (OSD, 2016) needs MINOR REVISION, the reasons are exposed below. My main concerns are the precalibration of the pH sensor, the adaptation to the high salinity conditions in the Western Mediterranean Sea and the comparison with ship based observations as detailed below.

SPECIFIC COMMENTS

1. Introduction

I do not understand why the acronyms for dissolved oxygen and dissolved inorganic carbon widely referred as $O_2$ and DIC among others in the ocean biogeochemistry literature are here referred as $c(O_2)$ and $c(DIC)$ .. why the "c"?. I think is unnecessary and could be misinterpreted.

The only thing I miss in the introduction is short paragraph about the particular CO2 chemistry characteristics in the Western Mediterranean Sea. Some information about this can be found in the literature, for example Rivaro et al. (Mar Chem, 2010), Touratier & Goyet (DSRI, 2011), Álvarez et al. (OcSc, 2014), Yao et al. (Mar. Envirn. Res., 2016) among others. The MedSea

is warm, salty, very high in alkalinity and high in pH compared to rest of the open ocean, concretely the adjacent Atlantic. This fact should be highlighted in my opinion.

2. Methodology ( I would change the title of this section to Data set and methodologies)

I would suggest a reorganization of this section. I think it can be improved and some more information should be given.

2.1. REP14-MED sea trial.

Despite the general overview of this trial is given in Onken et al (this issue). It might be good to just write a short paragraph about the general aim of deploying 11 gliders and concretely one with a pH ( and other biogeochemical ) sensors.

2.2 Glider sensors.

I am not an expert on the different types of ISFET sensors, so it is not clear to me if the glider had a $pCO_2$ sensor separated from a pH sensor, or is it a dual sensor?.

Please clearly separate the paragraphs according to the sensors described, first conductivity and temperature, then oxygen, then $pCO_2$ and finally pH. Give details about the quality control for each of them. For example no reference is given for the O2-glider calibration, I am sure this data is analysed in other manuscript, and if not please provide this information because the data is presented and discussed along with the final pH-glider data.

2.3. Ship based measurements ( I would change the current title "validation of ISFET pH measurements").

In this section I would also include for example a description of the oxygen winkler measurements if any done to calibrate the glider O2 sensor.

During the CO2 compiling exercise CARINA the Mehrbach et al. (1973) constants refitted by Dickson and Millero (1987) (see Key et al., 2010) were suggested to calculate pH from DIC and TA, as also concluded by Álvarez et al. (2014), specifically for the MedSea, however in GlodapV2 the ones used by Lueker et al. (2000) were used. Please comment about this.

Please clearly state that both pH from the ship and from the glider are expressed on the total scale and at in situ temperature.

3. Results and Discussion (the title "Results and corrections" is not very appropriate for a journal section).

I do not see a clear separation between the different sections included here, the pH corrections and validation are given in 3.1, 3.2 and 3.3. I suggest a reorganization of the whole section to make it more readable.

3.1. Ship based data (present and discuss the DIC, TA and pH derived ship based data)

TA and DIC measurements are expensive and time consuming, I think they deserve to be presented within a section from coast to open ocean ( and also the pHT in situ ) derived from them. Figure 4 a b and c are just showing vertical ranges of variability, but some coast-ocean variability should be also seen in the data. CTD hydrographic temperature and salinity data should also be included.

I do not understand the first paragraph in section 3.1... what do you mean with the standard deviation values, are you calculating bin averages and STD by depth intervals?.

Fig 4c is useless, are you showing pH total scale at in situ temperature?.. please adjust the pH range if you keep it.

3.2. Glider data

3.2.1. Sensors validation

First comment on the Temperature and salinity glider data referring to the vertical distribution in Fig 4 e and f.

Secondly some words and maybe a figure (now missing) about the comparison between ship and glider O2 data.

Finally all your findings about the pH glider data.

I would start commenting Fig 4d.

3.2.2. Coast to open ocean high resolution hydrographic and chemical variability from glider data

Here you should comment fig 8 & 9.

In general this section also needs some bibliography review. Please check the references given above that should be also included when discussing pH values and variability associated with distinct water masses in the MedSea.

4. Conclusions

Please the final phrase I think it should include "potential use of the corrected glider pH ....." as the pH sensors are still under development and in situ checking.

I hope to have been helpful.

---

## Referee Comment (RC2) · Anonymous Referee #2 · 16 Dec 2016

The manuscript by Hemming et al. describes a deployment of a seaglider in the Mediterranean Sea equipped with an experimental pH sensor. The performance of this pH sensor is evaluated against shipboard carbonate system parameter measurements of $A_T$ and DIC and the authors suggest a number of corrections to make their data fit their expectations. This is an instructive example of the challenges of implementing and field testing new sensor technology and represents one of the first reports on pH measurements from gliders.

I have some general comments:

The presentation of the data manipulations and suggested corrections appear very diagnostic and data-driven. At the same time, there are plausible physical causes to most of them, which the authors present, too (e.g., ambient light effect on FETs;

temperature effect on E*). It's likely mainly a question of style, but my preference would be to always start from the sensor knowledge to explain an observed mismatch and then suggest corrections, rather than an "our data didn't fit, so we made it fit" approach and then defending these corrections with theory after the fact.

More importantly, the goal of most (all?) the corrections seems to be to reduce the glider pH variability to the level seen in shipboard pH samples? This misses the point. Continuous, autonomous observations can very well be more variable than discrete measurements, in particular if the continuous measurement series captures time/spatial scales of variability or events that simply go undetected with coarser discrete sampling.
In that regard, the drift correction to pH(@14 °C) is a critical point that needs more detail and potentially a second look (details further below).

There is a lack of detail on the sensor used and its handling. This limits the utility and impact of the present study. Relevant information need to be added.

Specific comments:

P1L4: "Northwestern Mediterranean Sea" suggests a basin scale study and is maybe a bit a too generic description of the deployment location, i.e., a transect of just 100 km off the Sardinian coast?
Similarly, P1L14 and P1L16: "this region" is not well defined (I didn't know what it actually refers to), so I would suggest to closer specify the study region.

P2L9: Why not use ppm for the mole fraction?

P2L16: Potentially add the relevance of pH changes/anthropogenic $CO_2$ invasion to the study region/Northwestern Mediterranean?

P2L17: What is stochastic variability?

P3L11: Last sentence is irrelevant to the presented study.

P3 ISFET and glider sensors: What was the source of the sensor? Is it commercial/ semi-commercial/custom-built? Was the ISFET unit a commercial product (Honeywell?)? On what material support is it mounted (important to assess the pressure tolerance)? Is the packaging of the ISFET into a sensor a commercial/semi-commercial/ custom-built one?
This is essential information to put it into context of other studies with (other) ISFET pH sensors and directly affects the impact of this study.

P3 ISFET and glider sensors: The handling of the pH sensor needs to be described in detail. Was there any temperature or pressure compensation/calibration (in particular on $E^*$) other than described later in the manuscript? Were salinity/$Cl^-$ changes taken into account (as suggested by equation 1) for the calculations? How was the ISFET and the reference electrode stored before deployment: in NaCl solution, artificial seawater, Mediterranean seawater, at what salinity, how long before deployment? ...

P3L16: "the [other] retrieved data were of very poor quality". Any ideas why?

P4L18: I don't understand the figures. "DIC and $A_T$ differed by 3.1 and 2.5 $\mu$mol $kg^{-1}$, respectively" means that the second sample was always higher than the first one? I would hope that the difference between replicates would average around zero, otherwise this sounds like a serious methodological issue? I assume the authors refer either to the average absolute difference between replicates or the standard deviation between replicates?

P4 last sentence and first sentence on P5: This is unclear:

- How many casts were perfomed? (Should probably be mentioned in section 2.3 and/or P5L18)

- Why are there several standard deviations for a "standard deviation of the mean DIC/$A_T$ (averages over all casts)"?
  I kind of get the idea to split it into surface (top 150 m) and deep values, but

that only gives me two values. Instead, I see two ranges of standard deviations? Looking at the figures (4d), it seems like the data were aggregated into depth bins and – likely – the ranges are the numbers for the respective depth bins shallower than 150 m and deeper?

(a) This has to be explained in the text.

(b) Depending on the size of the depth bin, the depth gradient can become an important contributor to the standard deviation. Say all glider dives are identical, the standard deviation of the top 150 m would still be much higher than the bottom 150 m because of the higher depth variability near the surface compared to depth. Same for P5L9 and other statements like this (e.g., P7)

P5L14: What about the magnitude of diel variations? Because that's essentially what is looked at here.

P6L6-9: Is there experience from other autonomous deployments (floats?) in the literature that could be used?

Section 3.3 Correcting pH for drift, temperature, and pressure: Can you give more details about your corrections (equations, magnitude/values of m and c) to make it reproducible for others? Please also comment whether the temperature and pressure slopes are comparable to other findings (in particular P6L24 and Johnson et al. 2016)?

P6L27: "unrealistic scale" is unclear. Please specify or rephrase (large range?). In addition, scale is ambiguous here since it could refer to the different pH scales (total, seawater, ...).

P6L28/P6L30/P7L4/P1L8: What is it, a time-varying or a constant offset? Please be consistent to avoid confusion (or simply remove the constant in P6L30/P7L4?).

P6L28: "depth-constant" (uniform with depth) instead of "constant-depth" (applied to the same depth level)?

P6L30: The density gradient was weak, the pH gradient, too? You don't want to have

a gradient in your variable..
Can the depth of $\theta = 14\ ^\circ$C be made visible in one of the plots to get an idea of the depth range?

P6/P7 offset drift correction: How does the time evolution of the offset look like? Is it linear, exponential, or at least smooth (could be added to Figure 7)? If not, then what the authors measure is in fact not the pH but a pH anomaly relative to $pH_s(14\ ^\circ C)$, i.e., they remove the environmental variability of $pH_s(14\ ^\circ C)$ from their $pH_g$ data.

P7L5: If you derive the temperature correction from a subset with similar temperature gradients in the surface, is it applicable for the entire deployment/dives with different temperature gradients? Temperature certainly plays a role for these dives, too, but does it follow along the same relation? A look at figure 6 suggests that the selected stations cluster on one side of the corrected profiles, i.e., there is a bias? (Which might also cause some portion of the high surface variability in $pH_{gTPc}$?)

P7L13: And excluding daytime dives?

P7L14 vs. P7L21: in situ or potential temperature??

P7L28: "to achieve a match within the pH repeatability of the discrete samples"
That's not the point of continuous vs. discrete measurements. A higher variability in continuous data can easily be real.

P7L26: Indeed. Did you try any laboratory experiments with your pH sensor to confirm a temperature dependence (and salinity- and pressure dependence, if possible)? At least the temperature aspect should be easily feasible and would add significantly to solidify the correction approach.

P7L29: Can you comment on the uncertainty of your corrections and how that might affect your data? A linear temperature correction for ISFETs seems to be well-established, pressure corrections seem to be handled differently (e.g., this work, Johnson et al. 2016)?

P8L3: "at some locations": Imprecise, please specify (East/West/coastal/...?)

P8L5: Don't you have any data to support the DCM depth for your study? It seems like there were (at least) 12 gliders and two research vessels deployed..it should be possible to find (even an uncalibrated) Chlorophyll a fluorometer on a CTD among them..?

P8L17: "The spatial variability of these two regions differed for each time period" is unclear. Can you extend on this (what time periods; any relation of changing extend with displacement of isopycnals/water masses/SSHA)?

P8L18: "at a range of depths": Please specify. Were values similar along isopycnals E/W and the depth differences are just inclined density surfaces?

P8L23: Which time periods? (Maybe specify in section 2.1?)

P8L28: Sentence unclear to me. (Intrusion instead of encroachment?)

P8, section 3.4: This section describes the data and depth structure (first paragraph), it describes the East-West differences in the transect (second paragraph), and it discusses circulation aspects to explain mainly the physical oceanography data (third paragraph). What I think is missing in a section entitled "pH variability" is a biogeochemical discussion how to interpret the East West differences in pH. Is it related to a coastal/offshore gradient, to different preformed $pH/DIC/A_T/O_2$ concentrations in the respective water masses, to a gradient in nutrient supply and/or respiration (again: coastal/offshore gradient or likely water mass effect), ...? All these questions remain unanswered. (Potentially, part of the depth structure discussion of the first paragraph could be merged with this "fourth" paragraph.)

P9L4: Do you have any ideas/reason/speculation what caused the drift? The ISFET unit? E*? How could you reduce the drift in the first place or is it impossible to avoid?

P9L7-9: Again, a lab temperature study would solidify this result.

Fig 1: A distance scale in the left figure, too, would be nice.

Fig 1: What about the ca. 15 km North/South displacement between water samples and glider path for the match of water samples to glider dives?
I might have missed it, but did you describe in your methodology how you matched glider dives to ship hydrocasts? Shortest distance? Along equal longitude? The bathymetry suggests quite some difference at the same longitude close to the coast, so that a "distance from the coast" or "equal bottom depth" might be more adequate/give a better match?

Fig 4: What about a left/right grouping of water samples (DIC, $A_T$, $pH_s$; left top to bottom) and CTD/glider data ($\theta$, S, $pH_g$; right top to bottom)? This would avoid confusion about the legend next to 4c. Also, the legends could be placed inside the subpanels to gain some space (in particular to better see the subsurface maximum in $pH_s$)?

Fig 5: Maybe rename the y axis labels of panels b-e and the variables in the figure caption by $\Delta$X instead of X to emphasize the anomaly?

Fig 6: "offset drift correction" and 40 m?

(Fig 7: Make consistent with in situ / potential temperature of the correction description.)

Fig 8 and 9: Why did you split the plots into two figures? In my view, they would be more sensible as one (pH data together with its context). If space is a concern, you could think about removing the x axis labels and ticklabels for the upper panels since they are identical (as you did for the y axis labels and ticklabels for the center and right panels).

Minor: I would also appreciate a distinction between "the sensor"/"the ISFET sensor"/ "the ISFET pH sensor" and "the ISFET". The first refers to the ISFET including the packaging (housing, electronics, ...) the authors used (i.e., their experimental sensor)

[Figure]

while the second refers to the type of sensing probe (a transistor)/its working principle that can be shared by many different pH sensors but the one discussed here. It seems that in quite a few instances where "The ISFET ..." is used, it merely refers to "Our ISFET pH sensor ..." rather than to all ISFETs.

Typos:

P4L15: ...Scripps Insititution of Oceanography, USA, ...

P5L33: FET-based sensors

P7: "Tc" is sometimes italic and sometimes not

---

## Referee Comment (RC3) · Anonymous Referee #3 · 27 Dec 2016

This manuscript describes the deployment of an ISFET-pH sensor on a glider in the Mediterranean as part of the REP14-MED experiment. Shipboard carbonate measurements of dissolved inorganic carbon and total alkalinity were used to evaluate the sensor performance. The authors suggested a number of corrections to the ISFET data to make it fit with the shipboard measurements.

One of the major concerns I have with this paper, is that the author's main aim appears to be to reduce the variability of the glider samples to match the significantly lower resolution CTD samples. The much higher temporal resolution and greater sampling area of the glider will give greater variability in the pHg compared to the pHCTD. Therefore, I am concerned that the authors may be misguided in their application of corrections – perhaps the difference in resolution could be commented on and the corrections discussed further, or the data presented in such a way that the pHCTD measurements

are used as a guide rather than an elimination benchmark. This is discussed briefly in section 3.5 of Bresnahan et al., 2014. I understand that this correction of the sensor is based on the similarity of observed temperature and salinity measurements between CTD and glider – however, measurement techniques for these parameters are well established, with similar accuracy levels, and care should be taken when using the same standards for the ISFET pH sensor and pH calculated from bottled sampels.

The difference in variability could also be addressed with more information in the introduction on expected regional pH variability as seen from previous work in the Mediterranean (as briefly mentioned on page 5 line 14). This would demonstrate that temporal variability over the length of the deployment is minimal. Therefore, the procedures in the manuscript – correcting the data using 16 of the glider profiles, along with the pH of the bottled reference samples collected before the ISFET deployment time are valid for quality controlling the sensor.

Overall, I think this paper should be published with minor corrections. The manuscript gives an indication of the challenges when field testing new sensor technology, and is one of the first demonstrations of pH measurements a mobile platform.

Specific Points: P3 Section 2.2: More information on the ISFET-sensor used would be useful – specifically the calibration. It would also be interesting to know what the authors mean by poor quality –was this caused by integration into the glider electronics, or did the sensors malfunction? A brief sentence on this would also be useful – given that the paper is based around discussing challenges when field-testing sensors. The authors specify that they used a Cl-ISE. How long was this conditioned for? Previous studies (Bresnahan et al., 2014, Takeshita et al., 2014) both recommended conditioning in seawater levels of bromide ions before deployment to prevent reference electrode drifts.

What was the ionic strength of the two buffers used on deck to calibrate the ISFET? You also specify the pH of these solutions to a 4 decimal point (5 sig. figs). This is

very accurate for a pH sensor – particularly when the accuracy of the pH sensor you deploy is only 0.005. What pH system did you use to get this accurate buffer pH to calibrate your solutions? Was the deployed ISFET-measured pH of the buffer solutions the same before and after (i.e. was there any drift?)? Were the same solutions used – was there any drift in the solutions? Was there any noticeable biofouling on the ISFET sensor during the deployment?

Was there any lab-based temperature calibration done prior to deployment? Bresnahan et al., 2014 discuss a temperature error of <0.015 in their calibration of the sensors – this is greater than the specified accuracy of the deployed ISFET sensors. You mention the air temperature when calibrating with the buffer solutions, a measurement of the temperature of the buffer solutions would also be useful, particularly as you later correct for temperature dependence of the sensor. This is important, as the temperature of the solution may change the buffer pH (particularly when using such accurate pH figures) between the pre-deployment measurement and post-deployment measurement.

Finally, you provide a reference to Fukuba et al., 2008. This particular ISFET sensor does not have details of correction using buffers before and after deployment, but rather buffer solutions deployed with the sensor itself, allowing for in situ referencing. This is not the same procedure as the sentence is suggesting, nor does it provide an example of the converting the raw output to pH. Unless the ISFET sensor deployed had a similar "self-calibration" system, I would suggest removing this reference.

P4 Line 18: the difference in the DIC and the TA quoted from replicate samples – is this calculated from the standard deviation for each replicate? You state, in the previous sentence, there were two to three replicates collected per CTD cast – If this is not the standard deviation, how was this difference calculated between the three samples.

P4 Line 20: Please also state the borate-chlorinity ratio and the sulphate constants that were applied when using CO2SYS- with appropriate references. I realise these

may be quoted in the best practices section in the paper by Orr et al (2015), however it would be best if they were also specified here for clear understanding.

P4 Line 32: I find the range of standard deviations quoted throughout the manuscript to be confusing. For each specified bin (top 150m and below 150m) there is range of standard deviations quoted instead of one number for each bin. Is the standard deviation not calculated over the whole 150m? Is it further subdivided into smaller bins, and in which case what size are these bins and how many are there? I feel this should be clarified at the start of this section as the ranges are applied throughout the remainder of the manuscript. I assume these bins are the same as those specified in the caption for figure 5, but should be mentioned in the text for clarity.

P5 Line3: The authors refer to environmental variability when referring to the range of pH observed. This is not further discussed - What is the expected natural variability for the region? How much extra variability was observed and can be attributed to instrumental error? I realise that this is mentioned briefly in line 12, however numbers specifying the expected pH range and variability would be useful for those of us with little knowledge of the region. Furthermore, the instrumental error is not discussed in section 2.3. I think the authors meant sections 3.2 and 3.3.

P5 Line 22: Please specify if the same subtraction was performed on the salinity, dissolved oxygen and potential temperature.

P6 Line5: Does the ISFET have a constant offset caused by light? Or an offset changing with irradiance time/strength? Could you give some indication of the size of the offset based on your experiments.

P6 Line 28: I find it confusing when you discuss a constant depth –time varying offset, and then subsequently refer to, what I assume is the same correction, as a constant offset. It is not a constant offset as it varies with time. It also presumably varies with depth, as the correction was determined from the depth where the potential temperature was 14°C.

P7 Line 9: It would be good if the authors could specify the slope and the intercept of the linear regression in the text. This will allow better comparison with other studies.

P7 Line 27: The authors say poor-accuracy, is this relative to previous deployment? How did they determine the accuracy if the paper is based around correcting the pH sensor to the bottle samples? The best accuracy quotable for the sensor is that related to the reference samples.

P8 Line 7: Remove "there being"

Conclusions: The conclusion could be improved by summarising the findings of the paper including the biogeochemical variability (similar to the abstract). The authors also specify that the corrections they performed are not generally recommended or valid. A brief discussion of why these corrections are valid in this study, and under what other conditions they may not be valid would be good for future work by other studies.

Figures: (in general) seem to have a grey line around the edges. This is particularly on figure 8 where it looks like another figure was cropped out.

---

## Author Comment (AC1) · 25 Feb 2017

We thank the anonymous reviewer for making helpful suggestions on how to improve our manuscript.

NOTE: The original comments by the referee have been numbered 1-20, and red text has been used for the response by the authors.

1.  *I do not understand why the acronyms for dissolved oxygen and dissolved inorganic carbon widely referred as $O_2$ and DIC among others in the ocean biogeochemistry literature are here referred as $c(O_2)$ and c(DIC) .. why the "c"?. I think is unnecessary and could be misinterpreted.*

As referenced by Schwartz and Warneck (1995) – page 22, *'c' or 'C'* is the symbol used to represent a concentration of something. DIC and $O_2$ by themselves represent only the chemical species, hence the reason why we added the *'c'* before the brackets. This symbol has also been used by Castro-Morales and Kaiser (2012) published in *Ocean Science*. A few words explaining that *'c'* represents a concentration will be added to the manuscript for improved clarity.

2.  *The only thing I miss in the introduction is short paragraph about the particular CO2 chemistry characteristics in the Western Mediterranean Sea. Some information about this can be found in the literature, for example Rivaro et al. (Mar Chem, 2010), Touratier & Goyet (DSRI, 2011), Álvarez et al. (OcSc, 2014), Yao et al. (Mar. Envirn. Res., 2016) among others. The MedSea is warm, salty, very high in alkalinity and high in pH compared to rest of the open ocean, concretely the adjacent Atlantic. This fact should be highlighted in my opinion.*

A paragraph will be added to the manuscript's introduction to describe $CO_2$ chemistry characteristics, and particularly, the expected range of pH in the Western Mediterranean region, including the mentioned references.

3.  *I would suggest a reorganization of this section. I think it can be improved and some more information should be given.*

The following headings were suggested by the referee:

> *2. Methodology*
> *2.1 REP14-MED sea trial*
> *2.2 Glider sensors*
> *2.3 Ship based measurements*

We will reorganise and expand the methodology section as suggested, and we will give each sub-section the headings as suggested by the referee, to make it easier for readers.

4.  *Despite the general overview of this trial is given in Onken et al (this issue). It might be good to just write a short paragraph about the general aim of deploying 11 gliders and concretely one with a pH ( and other biogeochemical ) sensors.*

A short paragraph will be added to this section as suggested, describing the general aim of the REP14 campaign, and the context of the 11 glider deployment in which the glider trial was embedded.

5. *I am not an expert on the different types of ISFET sensors, so it is not clear to me if the glider had a pCO2 sensor separated from a pH sensor, or is it a dual sensor?.*

There were two dual pH/$p$($CO_2$) sensors on the glider. One sensor was integrated into the glider's electronics allowing the glider to control sampling, and one sensor was stand-alone. This will be more clearly explained in the updated manuscript.

6. *Please clearly separate the paragraphs according to the sensors described, first conductivity and temperature, then oxygen, then pCO2 and finally pH. Give details about the quality control for each of them. For example no reference is given for the O2-glider calibration, I am sure this data is analysed in other manuscript, and if not please provide this information because the data is presented and discussed along with the final pH-glider data.*

More information on quality control and sensor calibrations will be described (either in the relevant section, or as supplementary information), with text organised into paragraphs for each sensor in the order suggested by the referee.

7. *In this section [2.3 ship based measurements], I would also include for example a description of the oxygen winkler measurements if any done to calibrate the glider O2 sensor.*

Oxygen Winkler measurements were not used for the calibration of the glider's $c$($O_2$) sensor. Instead, the glider's oxygen optode was calibrated against measurements from a Seabird SBE 43 sensor deployed on the ship's CTD package. The method of calibrating the glider's $c$($O_2$) measurements will now be described.

8. *During the CO2 compiling exercise CARINA the Mehrbach et al. (1973) constants refitted by Dickson and Millero (1987) (see Key et al., 2010) were suggested to calculate pH from DIC and TA, as also concluded by Álvarez et al. (2014), specifically for the MedSea, however in GlodapV2 the ones used by Lueker et al. (2000) were used. Please comment about this.*

We used the lueker *et al.*, (2000) constants for the calculation of pH as these are the internationally recommended 'best-practice' ones (Dickson *et al*., 2007). However, the Mehrbach et al., (1973) refitted by Dickson and Millero (1987) constants will now be used, and a short sentence explaining our reasoning for using these constants, with reference to the CARINA exercise/Alvarez *et al*., (2014) will be added to the manuscript. The effect of this change on pH values is relatively small as the pH values derived using the Mehrbach et al., (1973) refitted by Dickson and Millero (1987) constants were on average 0.002 lower than pH derived using the lueker *et al.*, (2000) constants.

9. *Please clearly state that both pH from the ship and from the glider are expressed on the total scale and at in situ temperature.*

We previously stated that pH is on the total scale (e.g. P4L25), but the referee is correct in that we should clearly indicate that it is pH at *in situ* temperature. A few words stating this will be added to the manuscript.

10. *3. Results and Discussion (the title "Results and corrections" is not very appropriate for a journal section).*

We will use this title in the updated manuscript.

11. *I do not see a clear separation between the different sections included here, the pH corrections and validation are given in 3.1, 3.2 and 3.3. I suggest a reorganization of the whole section to make it more readable.*

We will reorganise the sections as suggested by the referee and we will give each section more informative headings.

12. *TA and DIC measurements are expensive and time consuming, I think they deserve to be presented within a section from coast to open ocean ( and also the pHT in situ ) derived from them. Figure 4 a b and c are just showing vertical ranges of variability, but some coast-ocean variability should be also seen in the data. CTD hydrographic temperature and salinity data should also be included.*

Yes, we agree that it would be useful to show transect sections of these parameters to better display spatial variability. We will include transects of $c$(DIC), $A_T$, and derived $pH_s$ from the ship samples, and also hydrographic sections of temperature, salinity, $c(O_2)$, and Fluorescence from the CTD in the updated version of the manuscript.

13. *I do not understand the first paragraph in section 3.1... what do you mean with the standard deviation values, are you calculating bin averages and STD by depth intervals?.*

Yes, this is exactly what we have done. We will clarify this better in the updated version of the manuscript.

14. *Fig 4c is useless, are you showing pH total scale at in situ temperature?.. please adjust the pH range if you keep it.*

Yes, we are showing this, and we chose this pH range on the x-axis for consistency when comparing with Fig. 4d. However, we agree that it would be more useful to decrease the pH range in order to see the variability in pH profiles as a function of depth more clearly. The pH range (x-axis) will be reduced and we will mention that pH is on a total scale at *in situ* temperature in the figure's caption.

15. *First comment on the Temperature and salinity glider data referring to the vertical distribution in Fig 4 e and f. Secondly some words and maybe a figure (now missing) about the comparison between ship and glider O2 data.*

We will show and discuss the comparison between ship and glider measurements of temperature, salinity, and $c(O_2)$ in the updated manuscript. We will create a new figure containing Fig. 4e-f, accompanied by a comparison plot of ship and glider $c(O_2)$ values, displayed in similar fashion.

16. *Finally all your findings about the pH glider data. I would start commenting Fig 4d.*

A discussion of our findings concerning the pH glider data (e.g. corrections, light effect) will be discussed together as proposed by the referee. We will start this discussion by commenting on the pH profiles

displayed in Fig. 4d.

*17.    Here you should comment fig 8 & 9.*

The referee is referring to a new proposed subsection (not copied here) based on ocean to coast glider data. We will discuss Fig. 8 & 9 in this proposed subsection.

*18.    In general this section also needs some bibliography review. Please check the references given above that should be also included when discussing pH values and variability associated with distinct water masses in the MedSea.*

The references from comment 8 above, as well as elsewhere, will be cited in this section when discussing variability in values of pH and the other parameters.

*19.    Please the final phrase I think it should include "potential use of the corrected glider pH ....." as the pH sensors are still under development and in situ checking.*

This sentence will be modified in the new version of the manuscript.

*20.    I hope to have been helpful.*

Thank you for taking the time to read through the manuscript. The comments have been very useful in improving the manuscript.

---

## Author Response (AR1)

| Pages 2 – 6:   | Review 1* |
|----------------|-----------|
| Pages 7 – 17:  | Review 2* |
| Pages 18 – 24: | Review 3* |

\* Includes the referees' comments (black) and author's response (red) as uploaded to Ocean Science on Feb 25th 2017, and the author's changes (blue).

| Pages 25 – 53: | Revised manuscript                     |
|----------------|----------------------------------------|
| Pages 54 – 84: | Revised manuscript with marked changes |

We thank the anonymous reviewer for the suggestions, which greatly helped to improve our manuscript.

NOTE: The original comments by the referee have been numbered 1-20, red text has been used for the response by the authors, and blue text has been used to describe the authors' changes in the manuscript. The page and line numbers refer to the version of the manuscript with tracked changes.

1. I do not understand why the acronyms for dissolved oxygen and dissolved inorganic carbon widely referred as O2 and DIC among others in the ocean biogeochemistry literature are here referred as c(O2) and c(DIC) .. why the "c"?. I think is unnecessary and could be misinterpreted.

As referenced by Schwartz and Warneck (1995) – page 22, 'c' or 'C' is the symbol used to represent a concentration of something. DIC and O2 by themselves represent only the chemical species, hence the reason why we added the 'c' before the brackets. This symbol has also been used by Castro-Morales and Kaiser (2012) published in *Ocean Science*. A few words explaining that 'c' represents a concentration will be added to the manuscript for improved clarity.

**"With 'c' representing a concentration" has been added to P2L6.**

2. The only thing I miss in the introduction is short paragraph about the particular CO2 chemistry characteristics in the Western Mediterranean Sea. Some information about this can be found in the literature, for example Rivaro et al. (Mar Chem, 2010), Touratier & Goyet (DSRI, 2011), Álvarez et al. (OcSc, 2014), Yao et al. (Mar. Envirn. Res., 2016) among others. The MedSea is warm, salty, very high in alkalinity and high in pH compared to rest of the open ocean, concretely the adjacent Atlantic. This fact should be highlighted in my opinion.

A paragraph will be added to the manuscript's introduction to describe CO2 chemistry characteristics, and particularly, the expected range of pH in the Western Mediterranean region, including the mentioned references.

Three paragraphs have been added to the introduction, starting on P2L28.

3. I would suggest a reorganization of this section. I think it can be improved and some more information should be given.

The following headings were suggested by the referee:

2. Methodology
 2.1 REP14-MED sea trial
 2.2 Glider sensors
 2.3 Ship based measurements

We will reorganise and expand the methodology section as suggested, and we will give each sub-section the headings as suggested by the referee, to make it easier for readers.

We have reorganised the paper as suggested.

4. Despite the general overview of this trial is given in Onken et al (this issue). It might be good to just write a short paragraph about the general aim of deploying 11 gliders and concretely one with a pH ( and other biogeochemical ) sensors.

A short paragraph will be added to this section as suggested, describing the general aim of the REP14 campaign, and the context of the 11 glider deployment in which the glider trial was embedded.

The general aims of the deployment have been added to Sect. 2.1.

5. I am not an expert on the different types of ISFET sensors, so it is not clear to me if the glider had a pCO2 sensor separated from a pH sensor, or is it a dual sensor?.

There were two dual pH/p(CO2) sensors on the glider. One sensor was integrated into the glider's electronics allowing the glider to control sampling, and one sensor was stand-alone. This will be more clearly explained in the updated manuscript.

This is now more clearly explained in the paragraph on P5L18.

6. Please clearly separate the paragraphs according to the sensors described, first conductivity and temperature, then oxygen, then pCO2 and finally pH. Give details about the quality control for each of them. For example no reference is given for the O2-glider calibration, I am sure this data is analysed in other manuscript, and if not please provide this information because the data is presented and discussed along with the final pH-glider data.

More information on quality control and sensor calibrations will be described (either in the relevant section, or as supplementary information), with text organised into paragraphs for each sensor in the order suggested by the referee.

This has been done in Sect. 2.2.

7. In this section [2.3 ship based measurements], I would also include for example a description of the oxygen winkler measurements if any done to calibrate the glider O2 sensor.

Oxygen Winkler measurements were not used for the calibration of the glider's c(O2) sensor. Instead, the glider's oxygen optode was calibrated against measurements from a Seabird SBE 43 sensor deployed on the ship's CTD package. The method of calibrating the glider's c(O2) measurements will now be described.

**This information has been added on P4L24.**

8. During the CO2 compiling exercise CARINA the Mehrbach et al. (1973) constants refitted by Dickson and Millero (1987) (see Key et al., 2010) were suggested to calculate pH from DIC and TA, as also concluded by Álvarez et al. (2014), specifically for the MedSea, however in GlodapV2 the ones used by Lueker et al. (2000) were used. Please comment about this.

We used the lueker *et al.*, (2000) constants for the calculation of pH as these are the internationally recommended 'best-practice' ones (Dickson *et al.*, 2007). However, the Mehrbach et al., (1973) refitted by Dickson and Millero (1987) constants will now be used, and a short sentence explaining our reasoning for using these constants, with reference to the CARINA exercise/Alvarez *et al.*, (2014) will be added to the manuscript. The effect of this change on pH values is relatively small as the pH values derived using the Mehrbach et al., (1973) refitted by Dickson and Millero (1987) constants were on average 0.002 lower than pH derived using the lueker *et al.*, (2000) constants.

9. Please clearly state that both pH from the ship and from the glider are expressed on the total scale and at in situ temperature.

We previously stated that pH is on the total scale (e.g. P4L25), but the referee is correct in that we should clearly indicate that it is pH at *in situ* temperature. A few words stating this will be added to the manuscript.

The sentence on P7L18 has been updated.

10. 3. Results and Discussion (the title "Results and corrections" is not very appropriate for a journal section).

We will use this title in the updated manuscript.

We have done this.

11. I do not see a clear separation between the different sections included here, the pH corrections and validation are given in 3.1, 3.2 and 3.3. I suggest a reorganization of the whole section to make it more readable.

We will reorganise the sections as suggested by the referee and we will give each section more informative headings.

Sect. 3 has been reorganised as:

3.1 Ship based data
3.2 Glider data
3.2.1 Temperature, salinity, and oxygen validation
3.2.2 ISFET pH validation
3.2.3 Coast to open ocean high resolution hydrographic and biogeochemical variability

12. TA and DIC measurements are expensive and time consuming, I think they deserve to be presented within a section from coast to open ocean ( and also the pHT in situ ) derived from them. Figure 4 a b and c are just showing vertical ranges of variability, but some coast-ocean variability should be also seen in the data. CTD hydrographic temperature and salinity data should also be included.

Yes, we agree that it would be useful to show transect sections of these parameters to better display spatial variability. We will include transects of c(DIC), AT, and derived pHs from the ship samples, and also hydrographic sections of temperature, salinity, c(O2), and Fluorescence from the CTD in the updated version of the manuscript.

These transects are now included in Fig. 4 and Fig. 5.

13. I do not understand the first paragraph in section 3.1... what do you mean with the standard deviation values, are you calculating bin averages and STD by depth intervals?.

Yes, this is exactly what we have done. We will clarify this better in the updated version of the manuscript.

A short paragraph explaining the procedure has been added to P7L21.

**14. Fig 4c is useless, are you showing pH total scale at in situ temperature?.. please adjust the pH range if you keep it.**

Yes, we are showing this, and we chose this pH range on the x-axis for consistency when comparing with Fig. 4d. However, we agree that it would be more useful to decrease the pH range in order to see the variability in pH profiles as a function of depth more clearly. The pH range (x-axis) will be reduced and we will mention that pH is on a total scale at *in situ* temperature in the figure's caption.

The pH range on the x-axis has been changed, now displayed in Fig. 5f.

15. First comment on the Temperature and salinity glider data referring to the vertical distribution in Fig 4 e and f. Secondly some words and maybe a figure (now missing) about the comparison between ship and glider O2 data.

We will show and discuss the comparison between ship and glider measurements of temperature, salinity, and c(O2) in the updated manuscript. We will create a new figure containing Fig. 4e-f, accompanied by a comparison plot of ship and glider c(O2) values, displayed in similar fashion.

This is now discussed in Sect. 3.2.1 at the beginning of the section on glider data results. Fig. 6 has been created to show glider vs. ship measurements for temperature, salinity, and *c*(O2).

**16. Finally all your findings about the pH glider data. I would start commenting Fig 4d.**

A discussion of our findings concerning the pH glider data (e.g. corrections, light effect) will be discussed together as proposed by the referee. We will start this discussion by commenting on the pH profiles displayed in Fig. 4d.

We start the discussion on the pH profiles at the beginning of Sect 3.2.2. The pH corrections and the light effect are discussed in the same section together.

17. Here you should comment fig 8 & 9.

The referee is referring to a new proposed subsection (not copied here) based on ocean to coast glider data. We will discuss Fig. 8 & 9 in this proposed subsection.

This has been done in Sect. 3.2.3. However, Fig. 8 and Fig. 9 are now combined to form Fig. 12.

18. In general this section also needs some bibliography review. Please check the references given above that should be also included when discussing pH values and variability associated with distinct water masses in the MedSea.

The references from comment 8 above, as well as elsewhere, will be cited in this section when discussing variability in values of pH and the other parameters.

**Bibliography review has been added to Sect. 3.2.3.**

19. Please the final phrase I think it should include "potential use of the corrected glider pH ....." as the pH sensors are still under development and in situ checking.

This sentence will be modified in the new version of the manuscript.

We have changed this sentence on P15L29.

20. I hope to have been helpful.

Thank you for taking the time to read through the manuscript. The comments have been very useful in improving the manuscript.

We thank the anonymous reviewer for the suggestions, which greatly helped to improve our manuscript.

NOTE: The original comments by the referee have been numbered 1-44, red text has been used for the response by the authors, and blue text has been used to describe the authors' changes in the manuscript. The page and line numbers refer to the version of the manuscript with tracked changes.

1. The presentation of the data manipulations and suggested corrections appear very diagnostic and data-driven. At the same time, there are plausible physical causes to most of them, which the authors present, too (e.g., ambient light effect on FETs; temperature effect on E\*). It's likely mainly a question of style, but my preference would be to always start from the sensor knowledge to explain an observed mismatch and then suggest corrections, rather than an "our data didn't fit, so we made it fit" approach and then defending these corrections with theory after the fact.

As the ISFET sensor was supposed to take into account temperature and pressure changes in the environment (Shitashima *et al.*, 2002; Shitashima, 2010), with a good level of accuracy (Shitashima *et al.*, 2013), ISFET pH measurements were not expected to differ significantly from the ship based pH measurements, particularly when considering past observations of pH in this region over a similar timescale (see comment 2. Below). We believe that the observed disagreement between ISFET sensor and ship-based pH data should be considered to be a result, which should be described first, followed by a discussion of the sensor, and lastly a proposed correction/course of action. We will make this logical progression clearer in the revised manuscript, by explaining the generally accepted views at the beginning, and pointing out where our results differ from this.

We have added explanations on P6L12 and on P10L33 to make the logical progression clearer.

2. More importantly, the goal of most (all?) the corrections seems to be to reduce the glider pH variability to the level seen in shipboard pH samples? This misses the point. Continuous, autonomous observations can very well be more variable than discrete measurements, in particular if the continuous measurement series captures time/ spatial scales of variability or events that simply go undetected with coarser discrete sampling. In that regard, the drift correction to pH(@14

 $^{\circ}$ *C) is a critical point that needs more detail and potentially a second look (details further below).*

We agree with the referee that variability seen by the glider should be different to ship-based measurements; this after all is the advantage of using gliders. We did see a larger variability in the higher resolution profiles from the glider, compared with ship measurements. However the magnitude of the variability seen in the glider-borne sensor was well outside the likely maximum range of pH variability seen in the literature (as well as the ship measurements in this study). This is particularly true when looking at measurements obtained at greater depths where we would not expect large changes in pH. For example, pH measurements (T = 25°C) presented by Alvarez *et al.*, (2014) taken from a hydrographic transect in the western Mediterranean Sea over a similar timescale to our deployment, varied between 7.87 and 7.93 at depths greater than 100 m, whereas pH measured by the glider's ISFET sensor (Fig. 4d) shows pH ranging between 7.97 and 8.18, which is roughly three times larger. This suggested that the measured range of the ISFET pH was incorrect (i.e. drifted) and required correcting.

A paragraph will be added to the introduction describing past observations of *in situ* high resolution pH in general (e.g. Hofmann *et al.*, 2011), and hydrographic measurements specifically from the northwest Mediterranean Sea (e.g. Alvarez *et al.*, 2014) in order to provide readers a background of typical pH variability.

The drift correction is discussed further in the response to comments 20 & 21.

3. There is a lack of detail on the sensor used and its handling. This limits the utility and impact of the present study. Relevant information need to be added.

More information will be added to the manuscript. This is detailed in the response to comments 9.1-9.4.

See response to comments 9.1-9.4.

4. P1L4: "Northwestern Mediterranean Sea" suggests a basin scale study and is maybe a bit a too generic description of the deployment location, i.e., a transect of just 100 km off the Sardinian coast? Similarly, P1L14 and P1L16: "this region" is not well defined (I didn't know what it actually refers to), so I would suggest to closer specify the study region.

We agree with the referee that 'Northwestern Mediterranean Sea' represents a larger area than that observed by the glider. Instead we now refer to the region as the 'Sardinian Sea'.

This has been done at a number of places, such as on P1L4, P1L15 and P1L17.

5. P2L9: Why not use ppm for the mole fraction?

The term 'ppm' is an ambiguous unit (Schwartz and Warneck, 1995) and we prefer to use ' $\mu$ mol mol-1' as this is more specific.

No changes have been made.

6. *P2L16:* Potentially add the relevance of pH changes/anthropogenic CO2 invasion to the study region/Northwestern Mediterranean?

Sentences discussing this will be added to the introduction.

A discussion has been added on P2L28.

7. P2L17: What is stochastic variability?

We will simplify this sentence for increased clarity.

This sentence has been altered, now on P2L10.

8. P3L11: Last sentence is irrelevant to the presented study.

This sentence will be removed from the manuscript.

This sentence has been removed.

9. P3 ISFET and glider sensors:9.1 What was the source of the sensor? Is it commercial/ semi-commercial/custom-built?

The sensor is currently under trial and not commercially available. The sensor was custom-built by Kiminori Shitashima (Tokyo University of Marine Science and Technology, Japan) based on an ISFET sensor from Hitachi ULSI Systems Co., Ltd. ten years ago. It has previously been used by Shitashima *et al.*, (2008) and Shitashima *et al.*, (2013).

See comment 9.5.

9.2 Was the ISFET unit a commercial product (Honey- well?)?

The ISFET unit was made by Kiminori Shitashima via special order.

9.3 On what material support is it mounted (important to assess the pressure tolerance)?

The housing of the unit was made from acrylic resin, and the ISFET and CL-ISE were moulded with epoxy resin in the housing.

9.4 Is the packaging of the ISFET into a sensor a commercial/semi-commercial/ custom-built one?

The housing of the unit was custom-built and is not commercially available.

9.5 This is essential information to put it into context of other studies with (other) ISFET pH sensors and directly affects the impact of this.

Information provided in the responses to comments 9.1-9.4 will be added to the manuscript.

Information has been added as a paragraph in Sect 2.2.

10. P3 ISFET and glider sensors: The handling of the pH sensor needs to be described in detail.

10.1 Was there any temperature or pressure compensation/calibration (in particular on E\*) other than described later in the manuscript?

No temperature and pressure calibrations/compensations other than the corrections specified in the manuscript were performed on the ISFET data. A sentence stating this will be added to the manuscript.

A sentence has been added on P6L13.

10.2 Were salinity/Cl- changes taken into account (as suggested by equation 1) for the calculations?

Yes, salinity changes were taken into account when undertaking the calculations. This will now be described in the manuscript.

A sentence has been added on P6L15 stating this.

10.3 How was the ISFET and the reference electrode stored before deployment: in NaCl solution, artificial seawater, Mediterranean seawater, at what salinity, how long before deployment? ...

The ISFET and reference electrode were stored in a bucket of seawater for an hour before the deployment of the glider. The salinity of this water was about 38.05. This information will be added to the manuscript.

This has been added on P5L29.

11. P3L16: "the [other] retrieved data were of very poor quality". Any ideas why?

The data retrieved from these sensors could not be used due to quality issues. It is unclear why there was a problem with measurements obtained by the stand-alone p(CO2) sensor. However, we think the regular on/off cycling of electricity to the integrated dual sensor in between sampling did not allow it to function properly. This information will be added to the manuscript. A sentence outlining the authors' recommendation to potentially improve the integrated dual sensor will also be added to the conclusion

section.

A paragraph describing the quality issues of these sensors has been added on P5L18 and a short paragraph has been added to the conclusion section on P15L11.

12. P4L18: I don't understand the figures. "DIC and AT differed by 3.1 and 2.5 μmolkg-1, respectively" means that the second sample was always higher than the first one? I would hope that the difference between replicates would average around zero, otherwise this sounds like a serious methodological issue? I assume the authors refer either to the average absolute difference between replicates or the standard deviation between replicates?

We understand that this could be confusing, and the referee is correct in assuming that we used the average absolute difference between replicates, with the value to the right of the ' $\pm$ ' symbol representing the standard deviation of these absolute differences. However, we now think it will be better to list the mean standard deviation of the replicate samples. These values will instead be listed and the method will be explained in the updated manuscript.

This change can be seen on P7L6.

- 13. P4 last sentence and first sentence on P5: This is unclear:
  - 13.1 How many casts were performed? (Should probably be mentioned in section 2.3 and/or P5L18)

We agree that this should be further clarified as 'casts 24-51' is not specific. We will mention in the text that eight casts were performed.

'Eight casts' has now been written when referring to casts 24-51 (e.g. P6L22). These casts are also labelled in Fig. 1, Fig. 4 and Fig. 5.

13.2 Why are there several standard deviations for a "standard deviation of the mean DIC/AT (averages over all casts)"? I kind of get the idea to split it into surface (top 150 m) and deep values, but that only gives me two values. Instead, I see two ranges of standard deviations? Looking at the figures (4d), it seems like the data were aggregated into depth bins and – likely – the ranges are the numbers for the respective depth bins shallower than 150 m and deeper? This has to be explained in the text.

This has in part been addressed in the response to Review 1 – comment 13.

To calculate these ranges of standard deviations, the values from all profiles of a given variable (e.g. DIC,AT,  $pH_{s...}$ ) were sorted into 10 m depth bins down to a maximum depth of 1000 m. The mean and standard deviation was calculated for each one of these 10 m bins using the assorted data within. This produced two arrays; 100 x mean values and 100 x corresponding standard deviation values between the surface and 1000 m depth. Thus, the quoted standard deviation ranges (e.g. for the top 150 m) were defined using the minimum and maximum standard deviation calculated from these bins within the depth range (e.g. 15 out of 100 binned standard deviations for 150 m). We will make sure to explain clearly what these standard deviation ranges represent, and how they were calculated, in the updated manuscript.

A paragraph has been added on P7L21.

13.3 Depending on the size of the depth bin, the depth gradient can become an important contributor to the standard deviation. Say all glider dives are identical, the standard deviation of the top 150 m would still be much higher than the bottom 150 m because of the higher depth variability near the surface compared to depth. Same for P5L9 and other statements like this (e.g., P7)

The size of the depth bins was chosen to take into account the vertical pH gradient. Hence, for example, for the top 150 m of the water column, there are 15 standard deviation values. It is for the reason highlighted by the referee that a range of standard deviation values was given, rather than one value for the entire selected depth range (e.g. top 150 m).

More information (including the size of the depth bins) will be described in the manuscript, as mentioned in the response to Review 1 – comment 12.

This has been written in the paragraph on P7L21, and information has been added to some figure captions (e.g. Fig.7).

14. P5L14: What about the magnitude of diel variations? Because that's essentially what is looked at here.

There is not much in the literature specifically considering the diel variability of pH within the top 1000 m of the water column close to this part of the Mediterranean Sea. However, we will add references here that describe diel variations in pH at other location (e.g. Hofmann *et al.*, 2011).

A couple of sentences have been added to P3L14.

15. P6L6-9: Is there experience from other autonomous deployments (floats?) in the literature that could be used?

The authors have not been able to find in the literature experience of correcting ISFET pH measurements for ambient light on a glider/float.

**No changes have been undertaken.**

16. Section 3.3 Correcting pH for drift, temperature, and pressure: Can you give more details about your corrections (equations, magnitude/values of m and c) to make it reproducible for others? Please also comment whether the temperature and pressure slopes are comparable to other findings (in particular P6L24 and Johnson et al. 2016)?

More information will be added to this section. This will include the offset equation, delta pH equation, and temperature and pressure correction equations, incorporating the calculated slope and intercept coefficient values. These will be compared with other findings, such as Johnson *et al.*, 2016.

The offset and delta pH equations, and a combined pressure-temperature equation has been added on P11-12, Eq. 3-5.

17. P6L27: "unrealistic scale" is unclear. Please specify or rephrase (large range?). In addition, scale is ambiguous here since it could refer to the different pH scales (total, seawater, ...).

This sentence will be re-written in a clearer way, and 'scale' will not be used.

This sentence has been modified on P11L16.

18. P6L28/P6L30/P7L4/P1L8: What is it, a time-varying or a constant offset? Please be consistent to avoid confusion (or simply remove the constant in P6L30/P7L4?).

It is a depth-constant time-varying offset, as highlighted by comment 19 below. This offset will now be consistently referred to as a 'depth-constant time-varying' offset.

This offset is now constantly referred to as a depth-constant time-varying offset throughout the manuscript.

19. P6L28: "depth-constant" (uniform with depth) instead of "constant-depth" (applied to the same depth level)?

We now refer to a 'depth-constant time-varying' offset. See comment 18 above.

See follow-up comment above.

20. P6L30: The density gradient was weak, the pH gradient, too? You don't want to have a gradient in your variable.. Can the depth of  $\vartheta = 14 \degree C$  be made visible in one of the plots to get an idea of the depth range?

The mean depth where offset values were calculated was 188 ( $\pm$ 105) m which is generally below the thermocline. The majority of these offset values were obtained using pH values below 100 m depth where pH gradients were weaker, but some offset values were calculated at depths between 75 and 100 m where pH gradients were greater (see Fig. 4d/Fig. 6). However, the relationship between the calculated offset values and variability in other parameters (e.g. salinity and dissolved oxygen) was insignificant (see comment 21 below), suggesting the calculated offsets were mostly representative of instrumental drift rather than physical and biogeochemical variability. We will add a few sentences explaining these points in the manuscript.

Sentences have been added on P11L25.

The depths where the offset values were calculated (i.e. where  $T = 14^{\circ}C$ ) will be displayed in the updated version of the manuscript.

These depths are now displayed as pale blue scatter points in Fig. 12d-f.

21. P6/P7 offset drift correction: How does the time evolution of the offset look like? Is it linear, exponential, or at least smooth (could be added to Figure 7)? If not, then what the authors measure is in fact not the pH but a pH anomaly relative to pHs(14 °C), i.e., they remove the environmental variability of pHs(14 °C) from their pHa data.

The time evolution of the offset is varying with time and is essentially a pH anomaly relative to  $pH_s(14 \ ^{\circ}C)$ . As highlighted by the referee, this could be interpreted as environmental variability. We agree that it would be good to show the time evolution of the offsets, and a figure displaying offset pH values, salinity, and dissolved oxygen concentration (at 14  $\ ^{\circ}C$ ) with time will be added to the manuscript. Weak relationships ( $r^2 = 0.2$ ) were found between variability in the pH offset values and variability in salinity and  $c(O_2)$ . The new figure and linear regression analysis suggests a small proportion of the variability in pH offset values can be attributed to changing environmental conditions, and that the calculated pH offset values are mostly representative of instrumental drift. Sentences discussing this new figure and the relationships between pH offset values and salinity, and  $c(O_2)$ , will be added to the manuscript. This is discussed on P11L25, and a new figure (fig. 9) has been added to the manuscript to show the time evolution of salinity,  $c(O_2)$ , and the pH offset where temperature is 14°C.

22. P7L5: If you derive the temperature correction from a subset with similar temperature gradients in the surface, is it applicable for the entire deployment/dives with different temperature gradients? Temperature certainly plays a role for these dives, too, but does it follow along the same relation? A look at figure 6 suggests that the selected stations cluster on one side of the corrected profiles, i.e., there is a bias? (Which might also cause some portion of the high surface variability in pHaTPc?)

The referee is right that the 'pHg TPc Sel.' data points were generally situated to the right of ' $\mu$ pHg TPc' within the top 100 m of the water column. We have decided to now use measurements from all dives (i.e. no subset) for the temperature and pressure corrections, with light affected measurements removed from the top 50 m of the water column during the day. We think this will be a more robust approach. Figures 6 & 7 will be updated to display all dives with slopes and coefficients, and the corresponding text will be modified.

We now do not use a subset, and all reference to this has been removed from the manuscript.

**23. P7L13: And excluding daytime dives?**

As mentioned in comment 22, pH data affected by light within the top 50 m will now be excluded from the correctional procedures. These excluded measurements will be scattered on Fig. 6 for reference, and a sentence explaining this exclusion will be added to the updated manuscript in the relevant section.

Data affected by light in the top 50 m have been removed and is scattered orange in Fig.11.

24. P7L14 vs. P7L21: in situ or potential temperature??

'Potential temperature' on P7L21 was a mistake. It will be changed to 'in situ temperature'.

We have now changed this to 'in situ temperature', now on P11L21.

25. P7L28: "to achieve a match within the pH repeatability of the discrete samples" That's not the point of continuous vs. discrete measurements. A higher variability in continuous data can easily be real.

We agree that a greater range in pH variability measured continuously could be real in some cases, as was presented at various locations by Hofmann *et al.*, 2011. However, as discussed in comment 2, the magnitude of the variability observed by the glider-borne sensor was well outside the likely maximum range of pH variability seen in the literature (as well as the ship measurements in this study). This was particularly clear when looking at deep measurements, as glider-borne measurements varied within a range three times larger than the range measured by ship during a past expedition in the Mediterranean Sea when looking at measurements collected on a similar timescale (Alvarez *et al.*, 2014).

A paragraph has been added on P3L3 describing typical pH variability in the region, as seen by Alvarez *et al.*, 2014.

26. P7L26: Indeed. Did you try any laboratory experiments with your pH sensor to confirm a temperature dependence (and salinity- and pressure dependence, if possible)? At least the temperature aspect should be easily feasible and would add significantly to solidify the correction approach.

We agree that this would improve our understanding of the ISFET sensor. However, it was not possible to test the ISFET sensors under laboratory conditions after the deployment window.

No changes were made.

27. P7L29: Can you comment on the uncertainty of your corrections and how that might affect your data? A linear temperature correction for ISFETs seems to be well- established, pressure corrections seem to be handled differently (e.g., this work, Johnson et al. 2016)?

A few sentences will be added to the manuscript commenting on the differences between our corrections and those used in other papers, such as Johnson *et al.*, 2016.

Sentences have been added on P13L3.

28. P8L3: "at some locations": Imprecise, please specify (East/West/coastal/...?)

The longitudinal ranges of these locations will be added to this sentence.

This was modified on P13L21.

29. P8L5: Don't you have any data to support the DCM depth for your study? It seems like there were (at least) 12 gliders and two research vessels deployed..it should be possible to find (even an uncalibrated) Chlorophyll a fluorometer on a CTD among them..?

Fluorescence was measured by the ship's CTD instrument, and an increase can be seen at the depths where oxygen and pH increases, supporting the notion that this is the DCM depth. Fluorescence measured by the ship will be described in the manuscript.

A fluorescence transect can now be seen in Fig. 4, and has been discussed in Sect. 3.1 and Sect. 3.2.3.

30. P8L17: "The spatial variability of these two regions differed for each time period" is unclear. Can you extend on this (what time periods; any relation of changing extend with displacement of isopycnals/water masses/SSHA)?

The reader will be referred to Fig. 9 for the specific time periods, and a statement commenting on the relation between pH spatial variability and changes in temperature and salinity (i.e. water mass properties), and isopycnals will be added to the manuscript.

The reader is now referred to the time periods stated in Fig.12 (e.g. P14L6). More references to water masses (e.g. LIW) have been added, relating pH to temperature and salinity and isopycnals in Sect. 3.2.3 (P13L17).

31. P8L18: "at a range of depths": Please specify. Were values similar along isopycnals E/W and the depth differences are just inclined density surfaces?

The depth will be specified as 'deeper than 100 m'. It seemed these parameters followed isopycnal surfaces at a range of points in time and space, which is particularly clear in the top 200 m. This will be discussed in the manuscript.

This is discussed on P13L27.

32. P8L23: Which time periods? (Maybe specify in section 2.1?)

Again, the reader will be referred to the time periods labelled in Fig. 9.

The reader is now referred to the time periods stated in Fig.12 (e.g. P14L6).

33. P8L28: Sentence unclear to me. (Intrusion instead of encroachment?)

We have replaced 'encroachment' with 'intrusion' as suggested.

34. P8, section 3.4: This section describes the data and depth structure (first paragraph), it describes the East-West differences in the transect (second paragraph), and it discusses circulation aspects to explain mainly the physical oceanography data (third paragraph). What I think is missing in a section entitled "pH variability" is a biogeochemical discussion how to interpret the East West differences in pH. Is it related to a coastal/offshore gradient, to different preformed pH/DIC/AT/O2

concentrations in the respective water masses, to a gradient in nutrient supply and/or respiration (again: coastal/offshore gradient or likely water mass effect), ...? All these questions remain unanswered. (Potentially, part of the depth structure discussion of the first paragraph could be merged with this "fourth" paragraph.)

This section will be reorganised as suggested by the referee, and a biogeochemical discussion will be added to the manuscript.

This section (now Sect. 3.2.3) now includes a more detailed biogeochemical discussion.

35. P9L4: Do you have any ideas/reason/speculation what caused the drift? The ISFET unit? E\*? How could you reduce the drift in the first place or is it impossible to avoid?

As the referee suggested, we can speculate that the drift was likely related to the interface potential between the two n type silicon parts (source and drain) being affected. However, as it was not possible to test the ISFET sensor for drift, and that many variables were involved, it is difficult to identify the true cause of the drift.

It is possible the drift may have been caused by the lack of proper conditioning before the deployment. The ISFET was switched on and left in a bucket of seawater for just one hour, contrary to some weeks as suggested by others (e.g. Bresnahan et al., 2014). Putting aside that our sensor differed from the Honeywell Durafet sensor described by Bresnahan et al., 2014, and that the salinity sensitivity of our ISFET sensor was small, perhaps more time was needed for the ISFET to stabilise in seawater prior to deployment.

To determine the true cause of the drift, in future two ISFET sensors should be tested in laboratory conditions within a bridge arrangement circuit to attempt to isolate possible factors contributing to drift. A discussion concerning the possible cause of the drift will be added to the manuscript.

A sentences has been added on P11L17, and in the conclusion section on P15L18.

*36. P9L7-9: Again, a lab temperature study would solidify this result.*

We agree with the referee that this would be useful, but a lab temperature test was unfortunately not possible.

No changes to the manuscript were made.

37. Fig 1: A distance scale in the left figure, too, would be nice.

A distance scale will be added to the left panel in Fig. 1.

This can now be seen in Fig. 1.

38. 1 Fig 1: What about the ca. 15 km North/South displacement between water samples and glider path for the match of water samples to glider dives? I might have missed it, but did you describe in your methodology how you matched glider dives to ship hydrocasts?

We calculated the offset,  $\Delta pH$ , and  $\Delta pH_{Tc}$ , using the mean profile of the ship pH measurements (' $\mu$  pHs' blue profile in Fig. 6). This was decided as the pHs standard deviations were relatively small when compared with standard deviation values of the ISFET glider data. Mean pHs is indicated in Sect. 3.3 on P6L29, P7L11, and in Fig. 3. However, this was not specified within the caption for Fig. 7 which may have caused confusion. Equations for  $\Delta pH$  and  $\Delta pH_{Tc}$  will be added to the manuscript, and the caption for Fig. 7 will be modified.

We have updated the caption for Fig. 10 (formally Fig.7), and we now include the equation for  $\Delta pH$  on P12 Eq.4, which is similar to that used to calculate  $\Delta pH_{Tc}$ . Step 4 on P12 refers to Eq. 4 when discussing the calculation of  $\Delta pH_{Tc}$ .

38.2 Shortest distance? Along equal longitude? The bathymetry suggests quite some difference at the same longitude close to the coast, so that a "distance from the coast" or "equal bottom depth" might be more adequate/give a better match?

We thank the referee for suggestions on how to match the ship hydrocasts with the glider's measurements, however we did not match the individual bottle casts with glider measurements as described in comment 38.1 above.

No changes were made.

39. Fig 4: What about a left/right grouping of water samples (DIC,  $A_T$ ,  $pH_s$ ; left top to bot- tom) and CTD/glider data ( $\vartheta$ , S,  $pH_g$ ; right top to bottom)? This would avoid confusion about the legend next to 4c. Also, the legends could be placed inside the subpanels to gain some space (in particular to better see the subsurface maximum in  $pH_s$ )?

We plan to split Fig. 4 into four separate figures:

- 1. c(DIC), AT, and derived pH
- 2. temperature, salinity, fluorescence, and c(O2) from the ship CTD measurements
- 3. The comparison between  $pH_g$  and  $pH_s$  measurements
- 4. Comparisons between glider and ship measurements of temperature, salinity, and  $c(O_2)$ .

We think splitting Fig. 4 into separate figures, and visualising the spatial variability better as transects will improve the manuscript. We will take the advice (i.e. saving space) of the referee on board when creating the figures.

These figures are now included in the updated manuscript (Fig. 4-7).

40. Fig 5: Maybe rename the y axis labels of panels b-e and the variables in the figure caption by  $\Delta X$  instead of X to emphasize the anomaly?

We agree this would be better. This figure will be updated.

Fig. 8 (formally Fig.5) has been updated.

41. Fig 6: "offset drift correction" and 40 m? (Fig 7: Make consistent with in situ / potential temperature of the correction description.)

The caption for Fig. 6 will be modified, and the description of the temperature correction will be consistent with Fig. 7.

The caption for Fig. 11 (formally Fig.6) has been updated, and is consistent with Fig. 10 (formally Fig.7).

42. Fig 8 and 9: Why did you split the plots into two figures? In my view, they would be more sensible as one (pH data together with its context). If space is a concern, you could think about removing the x axis labels and ticklabels for the upper panels since they are identical (as you did for the y axis labels and ticklabels for the center and right panels).

We will combine Fig. 8 and 9 as suggested.

Former Fig. 8-9 have been combined to form Fig. 12.

43. Minor: I would also appreciate a distinction between "the sensor"/"the ISFET sensor"/ "the ISFET pH sensor" and "the ISFET". The first refers to the ISFET including the packaging (housing, electronics, ...) the authors used (i.e., their experimental sensor) while the second refers to the type of sensing probe (a transistor)/its working principle that can be shared by many different pH sensors but the one discussed here. It seems that in quite a few instances where "The ISFET ..." is used, it merely refers to "Our ISFET pH sensor ..." rather than to all ISFETs.

A distinction will be made between "the sensor"/"the ISFET sensor"/ "the ISFET pH sensor" and "the ISFET".

We have read through the updated manuscript, making sure to not refer to the ISFET unit when describing the sensor.

44. Typos: P4L15: ...Scripps Insititution of Oceanography, USA, ... P5L33: FET-based sensors P7: "Tc" is sometimes italic and sometimes not

These typos will be fixed in the updated manuscript.

We have fixed these typos.

We thank the anonymous reviewer for the suggestions, which greatly helped to improve our manuscript.

NOTE: The original comments by the referee have been numbered 1-15, red text has been used for the response by the authors, and blue text has been used to describe the authors' changes in the manuscript. The page and line numbers refer to the version of the manuscript with tracked changes.

 One of the major concerns I have with this paper, is that the author's main aim appears to be to reduce the variability of the glider samples to match the significantly lower resolution CTD samples. The much higher temporal resolution and greater sampling area of the glider will give greater variability in the pHg compared to the pHCTD. Therefore, I am concerned that the authors may be misguided in their application of corrections – perhaps the difference in resolution could be commented on and the corrections discussed further, or the data presented in such a way that the pHCTD measurements are used as a guide rather than an elimination benchmark. This is discussed briefly in section 3.5 of Bresnahan et al., 2014. I understand that this correction of the sensor is based on the similarity of observed temperature and salinity measurements between CTD and glider – however, measurement techniques for these parameters are well established, with similar accuracy levels, and care should be taken when using the same standards for the ISFET pH sensor and pH calculated from bottled samples.

We agree with the referee that variability seen by the glider could be different to the Ship measurements. We did see a larger extent of variability in the higher resolution profiles from the glider, compared with ship measurements. However, by comparing our ISFET measurements with observations by ship in the literature (as well as the ship measurements in this study), the extent in which pH should be expected to range in the northwestern Mediterranean Sea on a similar timescale is significantly smaller. For example, pH measurements (T = 25°C) presented by Alvarez *et al.*, (2014) taken from a hydrographic transect in the western Mediterranean Sea over a similar timescale to our deployment, varied between 7.87 and 7.93 at depths greater than 100 m, whereas pH measured by the glider's ISFET sensor (Fig. 4d) ranged between 7.97 and 8.18, which is roughly three times larger. This suggested that the measured range of the ISFET pH was incorrect (i.e. drifted) and required correcting.

A paragraph will be added to the introduction describing past observations of *in situ* high resolution pH in general (e.g. Hofmann *et al.*, 2011), and hydrographic measurements specifically from the northwest Mediterranean Sea (e.g. Alvarez *et al.*, 2014) in order to provide readers a background of typical pH variability. We will make sure to comment on the difference in resolution in Sect. 3.

Two paragraphs have been added to the manuscript on P3L3. We have commented on the resolution on P10L22 in Sect. 3.2.2.

We agree with the referee that care should be taken when comparing temperature, salinity, and pH between the glider and the ship. However, we think this comparison is valid, as like with the sensors used to measure temperature and salinity, the accuracy of the Marianda VINDTA 3C and the ISFET pH sensor should have been similar ( $\pm$ 0.005 ISFET pH e.g. Shitashima et al., 2002, and Marianda VINDTA 3C instrument determined by CRMs, and precision of 0.003 pH). Furthermore, the sparser ship measurements of temperature and salinity were within standard deviations of the high resolution glider measurements. From this, we can then assume that water mass properties were similarly represented in both ship and glider measurements, and that similarly this should be expected with pH.

We are aware that comparing ship and glider measurements of temperature and salinity represents physical properties only, so we will also compare dissolved oxygen concentrations observed by the glider and by the ship.

Fig. 6 in the updated manuscript shows a comparison of dissolved oxygen concentrations obtained by the glider and the ship's CTD package. This is discussed in Sect. 3.2.1.

1.1 The difference in variability could also be addressed with more information in the introduction on expected regional pH variability as seen from previous work in the Mediterranean (as briefly mentioned on page 5 line 14). This would demonstrate that temporal variability over the length of the deployment is minimal. Therefore, the procedures in the manuscript – correcting the data using 16 of the glider profiles, along with the pH of the bottled reference samples collected before the ISFET deployment time are valid for quality controlling the sensor.

As mentioned in the response to comment 1, the expected pH variability will be described in the updated manuscript. Past observations suggest that the extent to which ISFET pH varied in this trial over a similar timescale was greater, and therefore suggests the foundation in which we based our corrections is valid.

**See comment 1.**

2. P3 Section 2.2: More information on the ISFET-sensor used would be useful – specifically the calibration.

More information of the ISFET sensor will be included in the manuscript, relating to the sensor itself and the calibration procedure. Some details that will be added to the manuscript can be found in the responses to the proceeding 9 comments, and in review 2 – comments 9.1-9.5.

Details have been added on P5L12.

2.1 It would also be interesting to know what the authors mean by poor quality –was this caused by integration into the glider electronics, or did the sensors malfunction? A brief sentence on this would also be useful – given that the paper is based around discussing challenges when field-testing sensors.

As discussed in Review 2 – comment 11, we think the regular on/off cycling of electricity to the integrated dual sensor in between sampling did not allow it to function properly. A few sentences explaining this will be added to the updated manuscript.

Sentences have been added on P5L22.

2.2 The authors specify that they used a Cl-ISE. How long was this conditioned for? Previous studies (Bresnahan et al., 2014, Takeshita et al., 2014) both recommended conditioning in seawater levels of bromide ions before deployment to prevent reference electrode drifts.

The ISFET and CI-ISE were stored in a bucket of seawater for an hour before the deployment of the glider. The salinity of this water was about 38.05. This information will be added to the manuscript.

This information has been added on P5L29.

2.3 What was the ionic strength of the two buffers used on deck to calibrate the ISFET?

The buffers were made up in synthetic seawater of S = 35, which would have an ionic strength of about 0.7 M. A sentence about this will be added to the manuscript.

We have added this information on P6L1.

2.4 You also specify the pH of these solutions to a 4 decimal point (5 sig. figs). This is very accurate for a pH sensor – particularly when the accuracy of the pH sensor you deploy is only 0.005. What pH system did you use to get this accurate buffer pH to calibrate your solutions?

The buffer solutions of AMP and TRIS were created following SOP 6 from Dickson et al., (2007) and the pH values of these buffer solutions were taken from this reference, assuming that the temperature of the solutions was 25°C, and the salinity = 35. We did not measure the pH of the solutions by any other means. We will reduce the decimal points of the pH values listed in the manuscript, and we will add this information about the buffer solutions to the manuscript.

More information has been added to the paragraph on P6L1. We now show the pH values with 2 decimal points.

2.5 Was the deployed ISFET-measured pH of the buffer solutions the same before and after (i.e. was there any drift?)? Were the same solutions used – was there any drift in the solutions?

The same buffer solution batches were used before and after the deployment. The ISFET measured values of the buffer solutions at the end of the deployment differed to those measured before the deployment. This drift was corrected for using the calibration data before and after the deployment. This information will be added to the updated manuscript.

This has been described in the paragraph on P6L1.

2.6 Was there any noticeable biofouling on the ISFET sensor during the deployment?

It was clear after an inspection of the glider and sensor that there was no biofouling. We will state this in the manuscript.

**A sentence has been added on P6L19.**

2.7 Was there any lab-based temperature calibration done prior to deployment? Bresnahan et al., 2014 discuss a temperature error of <0.015 in their calibration of the sensors – this is greater than the specified accuracy of the deployed ISFET sensors.

The ISFET sensor was supposed to take into account temperature and pressure changes in the environment (Shitashima *et al.*, 2002; Shitashima, 2010), hence no lab-based temperature calibration was performed.

**No changes were made. We discuss this on P6L12.**

2.8 You mention the air temperature when calibrating with the buffer solutions, a measurement of the temperature of the buffer solutions would also be useful, particularly as you later correct for temperature dependence of the sensor. This is important, as the temperature of the solution may change the buffer pH (particularly when using such accurate pH figures) between the predeployment measurement and post-deployment measurement.

We will list the recorded temperature ranges of the buffer solutions before and after the deployment in the manuscript and we will comment on the uncertainty relating to possible pH changes of the buffer solutions as a result of changing temperature.

The temperature and uncertainty associated with this are discussed on P6L5.

2.9 Finally, you provide a reference to Fukuba et al., 2008. This particular ISFET sensor does not have details of correction using buffers before and after deployment, but rather buffer solutions deployed with the sensor itself, allowing for in situ referencing. This is not the same procedure as the sentence is suggesting, nor does it provide an example of the converting the raw output to pH. Unless the ISFET sensor deployed had a similar "self-calibration" system, I would suggest removing this reference.

This reference will be removed from the manuscript.

This reference has been removed.

3. P4 Line 18: the difference in the DIC and the TA quoted from replicate samples – is this calculated from the standard deviation for each replicate? You state, in the previous sentence, there were two to three replicates collected per CTD cast – If this is not the standard deviation, how was this difference calculated between the three samples.

We calculated the mean absolute difference between replicate samples, with the value to the right of the ' $\pm$ ' symbol representing the standard deviation of these absolute differences. However, we now think it will be better to list the mean standard deviation of the replicate samples in the updated manuscript.

**This is discussed on P7L6.**

4. P4 Line 20: Please also state the borate-chlorinity ratio and the sulphate constants that were applied when using CO2SYS- with appropriate references. I realise these may be quoted in the best practices section in the paper by Orr et al (2015), however it would be best if they were also specified here for clear understanding.

The suggested ratio and constants will be stated in the updated manuscript.

**These are stated on P7L16.**

5. P4 Line 32: I find the range of standard deviations quoted throughout the manuscript to be confusing. For each specified bin (top 150m and below 150m) there is range of standard deviations quoted instead of one number for each bin. Is the standard deviation not calculated over the whole 150m? Is it further subdivided into smaller bins, and in which case what size are these bins and how many are there? I feel this should be clarified at the start of this section as the ranges are applied throughout the remainder of the manuscript. I assume these bins are the same as those specified in the caption for figure 5, but should be mentioned in the text for clarity.

To calculate these ranges of standard deviations, the values from all profiles of a given variable (e.g. DIC, $A_T$ ,  $pH_s$ ...) were sorted into 10 m depth bins down to a maximum depth of 1000 m. The mean and standard deviation was calculated for each one of these 10 m bins using the assorted data within. This produced two arrays; 100 x mean values and 100 x corresponding standard deviation values between the surface and 1000 m depth. Thus, the quoted standard deviation ranges (e.g. for the top 150 m) were defined using the minimum and maximum standard deviation calculated from these bins within the depth range (e.g. 15 out of 100 binned standard deviations for the top 150 m). We will make sure to explain clearly how these standard deviation ranges were calculated in the updated manuscript before such ranges are listed.

A paragraph has been added on P7L22.

6. P5 Line3: The authors refer to environmental variability when referring to the range of pH observed. This is not further discussed - What is the expected natural variability for the region? How much extra variability was observed and can be attributed to instrumental error? I realise that this is mentioned briefly in line 12, however numbers specifying the expected pH range and variability would be useful for those of us with little knowledge of the region.

As mentioned in comment 1 above, a paragraph will be added to the introduction describing past observations of *in situ* high resolution pH in general, and hydrographic pH measurements specifically from the Mediterranean Sea. A discussion of the comparison between the natural variability of past observations by ship and the ISFET measurements on a similar timescale of a few weeks will be added to the manuscript. As the pH derived from bottle samples in this deployment varied within a similar range to past observations of pH (roughly 0.1 pH) when considering all measurements between the surface and 1000 m depth, the difference between the standard deviations of glider measurements and ship measurements in this study could be used as an indication of the instrumental error. The instrumental error would therefore have been between 0.03 and 0.09 in the top 150 m of the water column, and between 0.005 and 0.045 beneath this.

**See comment 1.**

7. Furthermore, the instrumental error is not discussed in section 2.3. I think the authors meant sections 3.2 and 3.3.

We thank the referee for highlighting a possible mistake. However, the authors were referring to the instrumental error associated with obtaining c(DIC) and  $A_T$  samples using the VINDTA instrument, which would have in part contributed to the standard deviation values obtained from the pHS measurements. This is described on page 4, lines 18-19 in section 2.3. We will alter this sentence to make this clearer.

We have modified this sentence, now on P8L23.

8. P5 Line 22: Please specify if the same subtraction was performed on the salinity, dissolved oxygen and potential temperature.

This will be specified in the updated version of the manuscript.

This has now been specified on P9L30.

9. P6 Line5: Does the ISFET have a constant offset caused by light? Or an offset changing with irradiance time/strength? Could you give some indication of the size of the offset based on your experiments.

The offset depended on irradiance strength (i.e. the value changed depending on how close the sensor was from the light source, and the type of bulb used), and remained relatively constant when the light source was turned on. The offset was roughly between -4 and -6 x  $10^8$  counts and between -1 and -2 x $10^8$  counts when the LED and Halogen lights were used, respectively. A sentence will be added to this paragraph describing these observations.

A sentence has been added on P9L11.

10. P6 Line 28: I find it confusing when you discuss a constant depth –time varying offset, and then subsequently refer to, what I assume is the same correction, as a constant offset. It is not a constant offset as it varies with time. It also presumably varies with depth, as the correction was determined from the depth where the potential temperature was 14°C.

Offset values were derived using the difference between mean pHs and pHg where the temperature of the water was 14 °C, and the depth of this indeed varied throughout the time period of the deployment. However, the authors refer to the method in which the offset was applied. In other words, the offset applied to the glider data did not change with depth (now referred to as 'depth-constant', see Review 2 – comment 18), but changed in time (i.e. a different offset value was determined for each dive profile). This will be further clarified in the manuscript.

We also will not refer to the offset as just 'constant', as this is not correct as highlighted by the referee.

We have made sure to refer to the offset as 'depth-constant time-varying' throughout the manuscript.

11. P7 Line 9: It would be good if the authors could specify the slope and the intercept of the linear regression in the text. This will allow better comparison with other studies.

More information will be added to the section. This will include the offset equation, delta pH equation, and temperature and pressure correction equations, incorporating the calculated slope and intercept coefficient values. These will be compared with other findings, such as Johnson *et al.*, 2016.

The offset and delta pH equations, and a combined pressure-temperature equation has been added on P11-12, Eq. 3-5. Our findings are compared with Johnson *et al.*, (2016) on P13L3.

12. P7 Line 27: The authors say poor-accuracy, is this relative to previous deployment? How did they determine the accuracy if the paper is based around correcting the pH sensor to the bottle samples? The best accuracy quotable for the sensor is that related to the reference samples.

The authors were referring to the ship based reference samples when stating the poor-accuracy of the ISFET measurements. This will be clarified in the text.

We have specified that this is relative to pHs on P13L11.

13. P8 Line 7: Remove "there being"

This will be removed.

This has been removed.

14. Conclusions: The conclusion could be improved by summarising the findings of the paper including the biogeochemical variability (similar to the abstract). The authors also specify that the corrections they performed are not generally recommended or valid. A brief discussion of why these corrections are valid in this study, and under what other conditions they may not be valid would be good for future work by other studies.

The conclusions section will be expanded to include findings on the physical and biogeochemical variability, and we plan to discuss the points made by the referee regarding the corrections.

The biogeochemical variability is now summarised on P15L30. Some sentences relating to the drift correction have been added to step 2 on P15L16.

15. Figures: (in general) seem to have a grey line around the edges. This is particularly on figure 8 where it looks like another figure was cropped out.

This was caused when editing the plots and will be removed. The plots will now also be uploaded as PDFs for better quality.

The figure quality has been improved. Grey lines have been removed.

**Measuring pH variability using an experimental sensor on an underwater glider**

Michael P. Hemming1,2, Jan Kaiser1, Karen J. Heywood1, Dorothee C.E. Bakker1, Jacqueline Boutin2, Kiminori Shitashima3, Gareth Lee1, Oliver Legge1, and Reiner Onken4

[revised manuscript text omitted]

The Mediterranean Sea comprises of just 0.8% of the global oceanic surface, but is regarded as an important sink for anthropogenic carbon due to its physical and biogeochemical characteristics (Álvarez et al., 2014). Between 1995 and 2012, surface c(DIC) increased by 3 µmol kg-1 a-1 in the northwest Mediterranean sea, consistent with a rise in temperature of

30  $0.06 \,^{\circ}\text{C} \,^{a^{-1}}$ , and a decrease in pH of  $0.003 \,^{a^{-1}}$  (Yao et al., 2016). In contrast, pH in the neighbouring North Atlantic ocean decreased by just  $0.0017 \,^{a^{-1}}$  associated with an increase in c(DIC) of around 1.4 µmol kg-1a-1, and a temperature rise of  $0.01 \,^{\circ}\text{C} \,^{a^{-1}}$  (Bates et al., 2012). The greater potential of the Mediterranean Sea to store anthropogenic carbon can be explained by its higher alkalinity, warmer temperatures, and thus lower Revelle factor (Álvarez et al., 2014; Touratier and Goyet, 2011), when compared with other oceans, such as the North Atlantic.

pH in the Mediterranean Sea is typically higher than most other oceanic regions (Álvarez et al., 2014). pH on the total scale normalised to  $25^{\circ}$ C (pHT,25) collected by ship between 1998 and 1999 within the northwestern Mediterranean Sea varied between 7.92 and 8.04 at the surface, and between 7.9 and 7.93 at depths greater than 100 m (Copin-Montégut and Bégovic, 2002). When considering the Mediterranean Sea as a whole, pHT,25 obtained by ship in April 2011 varied between 7.98 and

5 8.02 at the surface, and between 7.88 and 7.96 at greater depths (Álvarez et al., 2014). The peak-to-peak amplitude of the pH annual cycle in the northwest Mediterranean Sea is typically 0.1, with maxima and minima found in spring and summer, respectively (Yao et al., 2016).

Measurements of pH with higher temporal resolution, such as those measured by *in situ* sensors, can vary greatly depending on their location and depth. Hofmann et al. (2011) presented results of 15 deployments using SeaFET pH sensors close to

- 10 the surface at a number of locations worldwide. They found pH could vary by as much as 1.1 in extreme environments, such as those obtained close to volcanic  $CO_2$  vents off the coast of Italy, but as little as 0.02 in open ocean areas, such as in the temperate eastern Pacific Ocean, over a time period of 30 days. Hofmann et al. (2011) were able to capture diel cycles in pH, with the most consistent variations found in coral reef locations. pH was at a maximum early evening and at a minimum in the morning, and had amplitudes of between 0.1 and 0.25, similar in range to other studies based in subtropical estuaries (Yates
- 15 et al., 2007).

20

Autonomous underwater gliders offer the possibility to observe the oceanic system with a greater level of detail on both temporal and spatial scales when compared with ship measurements (Eriksen et al., 2001). A low consumption of battery power and a great degree of manoeuvrability enable such vehicles to cover large areas and profile depths of up to 1000 m during missions that can last from weeks to months at a time. They are suitable platforms for a range of sensors, measuring both physical and biogeochemical parameters (Piterbarg et al., 2014; Queste et al., 2012).

This paper is a contribution to the special issue 'REP14 - MED: A Glider Fleet Experiment in a Limited Marine Area'. The main goal of this paper is to describe the trial of a novel ion sensitive field effect transistor (ISFET) pH sensor which was attached to an autonomous underwater glider in the northwest Mediterranean Sea during the REP14 - MED sea experiment. The secondary objective is to provide a method of correcting pH measured by this sensor, and to discuss the spatial and

25 temporal variability observed. The experiment, the glider sensors, including the ISFET sensor, and the method of validation, are described in Sect.2. The ship based data is presented in Sect. 3.1, and a comparison between ship and glider measurements is made in Sect. 3.2.1. The initial pH results and validation, the method of further correcting pH, and an artifactual light-induced effect are described in Sect. 3.2.2. Corrected pH measurements are analysed alongside other collected parameters in Sect. 3.2.3, and the paper's conclusions are provided in Sect. 4.

**30 2 Methodology**

**2.1 REP14 - MED sea trial**

This trial took place between 6th and 25th June 2014 in the northwest Mediterranean Sea off the coast of Sardinia, Italy (Fig. 1). This was part of the Environmental Knowledge and Operational Effectiveness (EKOE) research program led by the North

Atlantic Treaty Organisation (NATO) Centre for Maritime Research and Experimentation (CMRE), based in La Spezia, Italy. This was the 5th Recognised Environmental Picture (REP) trial, which was jointly conducted by two research vessels; the NRV *Alliance* and the RV *Planet*.

Eleven gliders with varying pressure tolerances were deployed during the trial, each making repeated west-east transects

- 5 separated roughly 0.13° latitudinally from one another within the REP14 MED observational domain. One of these gliders was operated by the University of East Anglia (UEA); an iRobot Seaglider model 1KA (SN 537) with an ogive fairing. All gliders were deployed to meet the objectives of the trial, such as to improve ocean forecasting techniques (e.g. model validation, evaluation of forecasting skill), to conduct a cost/benefit analysis of autonomous gliders, to analyse mesoscale and sub-mesoscale features, and to test new glider payloads. The latter objective was perhaps most relevant to the deployment of
- 10 the UEA glider. A more in-depth overview of the REP14 MED trial, its objectives, and the collected observational data, is described by Onken et al. (2017).

The UEA glider completed a total of 126 dives between 11th and 23rd June 2014. The first 24 dives did not record pH and the last 9 dives were very shallow, leaving 93 usable dives. Successive dives were approximately 2 to 4 km apart, descending to depths of up to 1000 m.

**15 2.2 Glider sensors**

Conductivity, pressure and *in situ* temperature measurements were obtained by the glider using a Seabird Scientific glider payload CTD sensor (Fig. 2). These measurements were then used to obtain potential temperature ( $\theta$ ) and practical salinity.

Dissolved oxygen concentrations ( $c(O_2)$ , where 'c' refers to a concentration) were measured using an Aanderaa 4330 oxygen optode sensor positioned towards the rear of the glider fairing (Fig. 2). The method of calibrating  $c(O_2)$  closely followed that

- 20 described by Binetti (2016), using the oxygen sensor-related engineering parameters TCPhase and CalPhase, which will be summarised here. The first step involved correcting  $c(O_2)$  to account for the response time ( $\tau$ ) of the sensor, as the diffusion of  $O_2$  across the silicon foil of the sensor is not an instantaneous process. Each oxygen sensor has a different  $\tau$ , which depends on the structure, thickness, age, and usage of the foil (McNeil and D'Asaro, 2014), and external environmental conditions such as temperature. An average  $\tau$  of 17 seconds was obtained using the method outlined by Binetti (2016) in Sect. 2.3.1.
- 25 After correcting TCPhase for  $\tau$ , glider TCPhase profiles were matched in time and space with pseudo-CalPhase profiles backcalculated from measurements of  $c(O_2)$  obtained by the ship Seabird Scientific SBE 43 sensor (CTD package) using the manufacturer's set of optode calibration equations. The relationship between the glider TCPhase and the ship pseudo CalPhase was established, and the calculated slope and offset coefficients were used to correct glider CalPhase, required for calibrating  $c(O_2)$  measurements. A comparison between the ship  $c(O_2)$  measurements and calibrated glider  $c(O_2)$  measurements is made
- 30 in Sect. 3.2.1.

Glider variables have been processed using an open-source MATLAB based toolbox (https://bitbucket.org/bastienqueste/ueaseaglider-toolbox/) in order to correct for differing timestamp allocations, sensor lags (Garau et al., 2011; Bittig et al., 2014), and to tune the hydrodynamical flight model (Frajka-Williams et al., 2011). Outliers outside of a specified range (e.g. 6 standard deviations) were flagged and not used for analysis, and glider profiles were smoothed using a Lowess low-pass filter with a span of 5 data points (<4 m range), which implements a local regression using weighted linear least squares and a 1st order polynomial linear model. Individual profiles were inspected afterwards to ensure that potentially correct data points were not removed.

- The ISFET pH sensor used in this study (Fig. 2) was custom-built by a working group led by Kiminori Shitashima at the 5 Tokyo University of Marine Science and Technology (previously the University of Kyushu), and is not commercially available. 5 The ISFET unit was housed in acrylic resin material. The ISFET unit and the reference chlorine ion selective electrode (Cl-ISE) 6 were moulded with epoxy resin in the custom-built housing. The ISFET pH unit was stand-alone, meaning that the sensor was 7 not integrated into any of the onboard glider electronics. The power source of the sensor was 10.5 V, supplied by three 3.5 V 7 Li-ion batteries.
- The glider also carried another ISFET pH sensor that was integrated into the glider electronics (Fig. 2), as well as two  $p(CO_2)$  sensors (Shitashima, 2010), one stand-alone and one integrated. The data retrieved from the integrated pH sensor, and the  $p(CO_2)$  sensors could not be used due to quality issues. We think the regular on/off cycling of power to the integrated dual pH- $p(CO_2)$  sensor between sampling did not allow it to function properly. In future, we would suggest the addition of backup batteries to supply power to the sensor between sampling. The cause of the problem with the stand-alone  $p(CO_2)$  unit 15 is unclear.

15 is unclear.

20

To measure pH, the activity of  $H^+$  ions is determined using the interface potential between the semiconducting ion sensing transistor coated with silicon dioxide (SiO2) and silicon nitride (Si3N4), and the Cl-ISE. The ISFET pH sensor was previously found to have a response time of a few seconds with an accuracy of 0.005 pH, with suitable temperature and pressure sensitivities (Shitashima et al., 2002; Shitashima, 2010; Shitashima et al., 2013). Before deploying the sensor, the ISFET and Cl-ISE were conditioned (as recommended by Bresnahan et al. (2014) and Takeshita et al. (2014)) in a bucket of local sea surface water with a salinity of 38.05. However, due to time constraints, conditioning took place over just one hour, rather than weeks as specified by Bresnahan et al. (2014) and Takeshita et al. (2014). During the deployment, pH measurements were obtained every 1 to 2 m vertically.

- Measurements obtained by the ISFET pH sensor were converted from raw output counts to pH on a total scale using a two-point calibration with 2-aminopyridine (AMP) and 2-amino-2-hydroxymethil-1, 3-propanediol (TRIS) buffer solution before and after the deployment of the glider. The same buffer solutions (Wako Pure Chemical Industries Ltd.) created in synthetic seawater (S = 35, ionic strength of around 0.7 M) were used before and after deployment. These buffer solutions had a pH of  $6.79 \pm 0.03$  (AMP) and  $8.09 \pm 0.03$  (TRIS). The pH uncertainty of the buffer solutions takes into account the effect of changing air temperature, ranging between 30.5 and 33.3 °C during pre-calibration, and between 27.5 and 28 °C
- 30 during post-calibration. A linear fit using the raw output measured from these buffer solutions was used to convert the raw counts to pH (Shitashima et al., 2002). A drift was observed between the pH of these buffer solutions before and after the deployment, which was corrected for. As the ISFET sensor was previously described to have pressure-resistant performance and good temperature characteristics for oceanographic use (Shitashima et al., 2002; Shitashima, 2010), no compensations for temperature and pressure were performed on ISFET measurements at this stage. The ISFET pH sensor has a salinity sensitivity

 $\partial pH / \partial S = 0.011$  which was taken into account. The effect of biofouling on ISFET pH measurements, as well as on all other glider measurements, was ruled out after a post deployment inspection of sensors indicated no problems.

**2.3 Ship based measurements**

As the in situ ISFET pH sensor was under trial, some form of validation of the results was required. In total, 124 water samples

- 5 were collected from Niskin bottles sampled at 12 depths (down to 1000 m) using a CTD rosette platform at eight locations (eight casts, numbered 24 51) close to the path of the glider (Fig. 1). Water samples were collected between 05:19 Local Time (LT, UTC+2) on the 9th June and 16:58 LT on the 11th June. The glider ISFET pH sensor started operating at 16:36 LT on 11th June. Overall, measurements obtained by the glider and the CTD overlapped better in space than in time (Fig. 3).
- When collecting carbon samples, water was drawn into 250 mL borosilicate glass bottles from Niskin bottles on the CTD 10 rosette using tygon tubing. Bottles were rinsed twice before filling and were overflowed for 20 seconds, allowing the bottle volume to be flushed twice. Each sample was poisoned with 50  $\mu$ L of saturated mercuric chloride and then sealed using greased stoppers, secured with elastic bands and stored in the dark (Dickson et al., 2007). The total alkalinity ( $A_T$ ) and the c(DIC) of each water sample was measured in the laboratory using a Marianda Versatile INstrument for the Determination of Titration Alkalinity (VINDTA 3C, www.marianda.com). c(DIC) was measured by coulometry (Johnson et al., 1985) following standard
- 15 operating procedure SOP 2, and AT was measured by potentiometric titration (Mintrop et al., 2000) following SOP 3b, both described by Dickson et al. (2007). During the analytical process, 21 bottles of certified reference material (CRM, batch 107) supplied by the Scripps Institution of Oceanography, USA were run through the instrument to keep a track of stability and to calibrate the instrument. For each day in the lab, 1 CRM was used before and after the samples were processed. A total of 19 concurrent replicate depth water samples were collected, with around 2 to 3 replicates per CTD cast. Calculating the mean
- standard deviation of these replicate samples enabled a measure of the instrument precision. The mean standard deviation of the c(DIC) and  $A_{\text{T}}$  replicates was 1.7 µmolkg-1 and 1.4 µmolkg-1, respectively. This corresponds to a pH uncertainty of 0.003 for c(DIC) and  $A_{\text{T}}$ , respectively, resulting in a combined uncertainty of 0.009.

Once  $A_T$  and c(DIC) were known, pH could be derived using the CO2SYS program (Van Heuven et al., 2011). This calculation has an estimated pH probable error of around 0.006 due to uncertainty in the dissociation constants  $pK_1$  and  $pK_2$  (Millero,

- 25 1995). Temperature and salinity were obtained from the Seabird CTD sensor on the ship rosette sampler, and the seawater equilibrium constants presented by Mehrbach et al. (1973) were used as refitted by Dickson and Millero (1987), which has been recommended by previous studies (e.g. CARINA Data Synthesis Project) for the Mediterranean Sea (Álvarez et al., 2014; Key et al., 2010). The sulfate constant described by Dickson (1990), and the parameterisation of total borate presented by Uppström (1974), was used. More information on the equilibrium constants used in CO2SYS and other available carbonate
- 30 system packages is described by Orr et al. (2015). pH derived from water samples collected by ship and glider retrieved ISFET pH are both on the total pH scale (as described by Dickson (1984)) at *in situ* temperature, and will from now on be referred to as pHs and pHg, respectively.

Standard deviations ranges will from this point on be listed in this paper when referring to variability in measurements, such as  $pH_s$  and  $pH_g$ . To obtain these standard deviations ranges, data points for a given variable were sorted into 10 m depth bins down to a maximum depth of 1000 m. The standard deviation was then calculated for each bin.

**3 Results and discussion**

**5 3.1 Ship based data**

Measurements obtained by the ship CTD package provide an overview of the temporal and spatial variability at the time when water samples used to derive  $pH_s$  were collected (Fig. 4). The  $\theta$  gradient was strong in the top 100 m of the water column due to limited vertical mixing, with a maximum of between 19 and 23 °C found in the upper 10 m of the water column, decreasing to between 13 and 14 °C at depths greater than 100 m. The salinity was low in the top 100 m, increasing to a maximum at

- 10 around 400 to 600 m. These fresher waters in the top 100 m are likely modified Atlantic water (MAW), typically having a salinity of between 38 and 38.3 in the northwest Mediterranean Sea (Millot, 1999). These waters enter from the Atlantic Ocean through the Strait of Gibraltar, flowing along the North African coast. Some water makes its way northwards and follows the shelf back west towards the Atlantic Ocean (Rivaro et al., 2010; Millot, 1999). At deeper depths, warmer saltier waters were found east of 7.5° E, which is likely to be Levantine Intermediate Water (LIW), identified in the western Mediterranean Sea
- 15 by a salinity range of 38.45 and 38.65, and θ of between 13.07 and 13.88 °C (Rivaro et al., 2010), typically found at depths of between 200 and 800 m close to the shelf slope (Millot, 1999). c(O2) maxima were found at depths of between 20 and 90 m. The Mediterranean Sea on the whole is considered to be oligotrophic (Álvarez et al., 2014). However, a Deep Chlorophyll Maximum (DCM) is common at these depths when waters are thermally stratified (Estrada, 1996). There is a build-up of actively growing biomass with greater cell pigment content as a result of photoacclimation, due to increased concentrations of
- 20 nitrate, phosphate, and silicate, as well as sufficient levels of light at these depths (Estrada, 1996). It is likely this high  $c(O_2)$  was related to the DCM, further evidenced by the high chlorophyll fluorescence layer observed at 60 to 100 m depth, particularly in the east.

The objective of deriving pHs using  $A_T$  and c(DIC) was to make a comparison with pHg measured by the ISFET sensor. c(DIC) and  $A_T$  were greatest at depths below 250 m, with lower values seen closer to the surface (Fig. 5a-b), which is typical

- of the northwest Mediterranean Sea (Copin-Montégut and Bégovic, 2002; Álvarez et al., 2014). The higher values of  $A_{\rm T}$  and c(DIC) at depth and in the east support the notion that this is LIW, as this water mass has previously been identified as having an  $A_{\rm T}$  of around 2590 µmolkg-1 and c(DIC) of roughly 2330 µmolkg-1 (Álvarez et al., 2014), coinciding with the warmer, saltier waters. Mean c(DIC) and  $A_{\rm T}$  (averages over eight casts) have standard deviations of 6.1 to 11.9 µmolkg-1 and 5.9 to 10.6 µmolkg-1, respectively, for the top 150 m of the water column, and 1.7 to 3.9 µmolkg-1 and 3.7 to 7.6 µmolkg-1 for deeper
- 30 waters, respectively. pHs had a maximum of 8.14 between 50 and 70 m depth (Fig. 5c). Mean pHs have standard deviations of 0.004 to 0.011 within the top 150 m and 0.006 to 0.017 deeper than this. A proportion of these standard deviations can be explained by the instrumental error associated with the analysis of c(DIC) and  $A_T$  discussed in Sect. 2.3.

**3.2 Glider data**

**3.2.1 Temperature, salinity, and oxygen validation**

Since the sensors were calibrated before deployment, it was expected that the measurements from the glider would match those from the CTD, because any discrepancies between data sets would indicate possible instrumental or methodological issues with

- 5 the glider measurements. Mean profiles of  $\theta$ , salinity, and  $c(O_2)$  collected by the glider and by ship (Fig. 6) agreed well. Values obtained by both ship and glider were mostly within one standard deviation of one another. Mean  $\theta$  and salinity retrieved during the eight ship pHs casts differed from the binned mean calculated using all available REP14 - MED ship casts at depths between 100 and 500 m. However, this is likely related to temporal or spatial variability as mean  $\theta$  and salinity were within the range of all available glider measurements. Furthermore, differences of roughly 0.1 °C, 0.02, and 1.5 µmolkg-1 can be seen
- 10 for  $\theta$ , salinity, and  $c(O_2)$ , respectively, between the binned mean profile of CTD measurements and the binned mean profile of glider measurements at depths greater than 500 m. These differences in  $\theta$ , salinity, and  $c(O_2)$  are related to the different spatial distribution of the two datasets, as the glider measured predominantly at 40° N where deep cooler, fresher, waters were observed in the west (Fig. 4a-b), uncommon in other areas of the observational domain (Knoll et al., 2015b).

**3.2.2 ISFET pH validation**

- 15 Mean  $pH_g$  and  $pH_s$  agreed best between 60 and 250 m (Fig. 7), although  $pH_g$  variability was a lot higher than for  $pH_s$ . Larger differences between these profiles can be seen above and below this depth range, with  $pH_g$  0.1 higher at the surface and roughly 0.07 lower between 950 and 1000 m when compared with  $pH_s$ . The  $pH_s$  maximum at approximately 50 to 70 m depth was not apparent in the  $pH_g$  profile, with highest  $pH_g$  seen at the surface. The standard deviations for  $pH_g$  were large, between 0.044 and 0.114 in the top 150 m of the water column and between 0.027 and 0.053 at other points in the water column. Comparing
- all  $pH_g$  dive profiles obtained during the mission suggests a great degree of temporal and spatial variability, with pH ranging from 8.02 to 8.28 at the surface, and between 7.97 and 8.13 at 800 m depth.

A diel cycle in  $pH_g$  anomalies (calculated by subtracting the all time mean from the hourly means within a given depth interval) was found predominantly at depths shallower than 20 m (Fig. 8b). Lower pH was found between 09:00 and 18:00 LT, decreasing by > 0.1 between 12:00 and 14:00 LT. Contrastingly, potential temperature, salinity, and  $c(O_2)$  anomalies (calculated

- in the same way as  $pH_g$  anomalies) did not have strong diel cycles (Fig. 8c-e), suggesting that the decrease in pH was not caused by changing environmental conditions. Particularly, one might expect  $c(O_2)$  to have a similar pattern to pH if it were related to photosynthesis/respiration due to variations in  $p(CO_2)$  (Cornwall et al., 2013; Copin-Montégut and Bégovic, 2002). However,  $c(O_2)$  remained relatively constant throughout the day at all depth ranges implying that the level of biological activity in the Sardinian Sea did not change on average throughout the day and hence would not have caused this reduction in pHg.
- The decrease in pHg coincided with increased levels of solar irradiance (Fig. 8a) recorded at meteorological buoy M1 (Fig. 1) during the day at the surface, hence it was likely a light-induced instrumental artefact. The effect of light on the voltage output of FET-based sensors using SiO2 and Si3N4 sensitive layers is known (Wlodarski et al., 1986), as the presence of photons can excite electrons in the valence band of the semiconductor material, creating holes and allowing the flow of electrons to

the conduction band. This increases the voltage threshold, falsely measuring higher hydrogen ion activity, leading to lower apparent pH (Liao et al., 1999).

The effect of light on our sensor was investigated further by exposing two ISFET pH sensors to artificial light whilst placed in reference buffer solutions (TRIS and AMP) under laboratory conditions. The results (not shown here) confirmed that our ISFET sensor is affected by light. The light-induced offset depended on the strength and type of the light source, and which sensor was being used. The offset remained relatively constant whilst the light was turned on. A maximum pH offset of -0.7 ( $-6 \times 10^6$  counts) and -0.15 ( $-3 \times 10^6$  counts) was found when the LED and halogen lights were used, respectively.

There were not enough dives for a robust light correction, and an irradiance measuring sensor was not attached to the glider, hence data collected within the top 50 m between 05:00 and 21:00 LT, representing roughly 5 % of all  $pH_g$  measurements, were

10 not used in later analysis. In order to reduce this light effect on pH measurements in future, ISFET sensors will have to be placed on the underside of the glider or equipped with a light shield.

Comparing  $pH_g$  to  $pH_s$  indicated that the range observed by the ISFET sensor was much larger. It could be argued that this difference in range is due to the differing temporal and spatial resolution between the glider and the ship measurements. However, comparing  $pH_g$  further with pH measurements in the literature on a similar time and spatial scale (Álvarez et al.,

- 15 2014) suggests that this is not an issue with resolution.  $pH_{T,25}$  collected in the western Mediterranean Sea over a period of around 2 weeks (Álvarez et al., 2014), comparable in length to this trial, varied by roughly 0.02 at the surface and by around 0.08 at depths greater than 100 m. The range observed by the glider ISFET sensor was therefore thirteen times larger at the surface, and roughly 3 times larger at depths below 100 m. This difference in range cannot be explained by the high sampling frequency of the glider. Furthermore, the larger variations in pHg were not a result of changing environmental conditions, as
- 20 evidenced by the relatively stable  $c(O_2)$ ,  $\theta$ , and salinity measured by the glider, discussed in Sect. 3.2.1 and Sect. 2.3.3. It is likely that the ISFET pH measurements were not only related to the amount of hydrogen ion activity in the water, but also to the temperature and pressure that the sensor experienced, which was unexpected considering the sensor has previously shown good temperature and pressure characteristics (Shitashima et al., 2002; Shitashima, 2010; Shitashima et al., 2013). Furthermore, comparing ISFET measurements with pHs and pH presented by Álvarez et al. (2014) suggests that the accuracy
- of the sensor was not as good as that previously claimed (Shitashima et al., 2002). Therefore, it was necessary to correct  $pH_g$  measurements for instrumental drift, temperature, and pressure.

The response of the ISFET sensor can be described by the Nernst equation (Eq. (1)), which relates sensor voltage to hydrogen ion activity:

$$E = E^* - m_N \lg(a(\mathrm{H}^+)a(\mathrm{Cl}^-))$$
(1)

30 which incorporates the Nernst slope (Eq. (2)):

$$m_N = RT \ln(10)/F$$

5

(2)

where *T* is temperature (k), *R* is the gas constant (8.3145 J K-1 mol-1), *F* is the Faraday constant (96485 C mol-1),  $a(H^+)$  and  $a(Cl^-)$  are the proton and chloride ion activities, *E* is the measured voltage by the sensor (i.e. electromotive force), and *E*\* is representative of the two half-cells in the ISFET sensor forming a circuit (i.e. interface potential) (Martz et al., 2010). It is known that temperature and pressure have an effect on *E*\* (strong linear relationship), and that the Nernst slope is a function of

5 temperature. Also studies have shown that it is possible for ISFET sensors to experience some form of hysteresis as a result of changing T and pressure (Martz et al., 2010; Bresnahan et al., 2014; Johnson et al., 2016).

The first step in correcting  $pH_g$  aimed to reduce the measured extent of variability to within the measured limits of  $pH_s$ . This in-part removed the non-monotonous instrumental drift experienced by the sensor, which we think was likely due to the  $E^*$  between the two n type silicon parts of the semiconductor being affected. A depth-constant time-varying offset correction

10 (i.e. one offset value determined for each dive, applied to the entire profile) was applied (Eq. (3)) using the difference between mean  $pH_s$  and each  $pH_g$  dive measurement where *in situ* temperature was 14.0 °C, as water with this temperature was situated at a depth below the thermocline for most dives, where the density gradient was weak.

$$pH_{Offset} = pH_s(T),_{mean} - pH_g(T) \quad \text{for } T = 14 \pm 0.1 \,^{\circ}\text{C} \tag{3}$$

The calculated offset values as a function of time were compared with salinity and c(O2) where *in situ* temperature was constant at 14 °C (Fig. 9). Variability in salinity and c(O2) with time were strongly related (r2 = 0.97), whereas the relationship between pH offset values and salinity, and c(O2), were not (r2 of around 0.2). Furthermore, the majority of offset values were calculated below 100 m (Fig. 12d-f), where the density and pH gradients were weak. This indicated that our depth-constant time-varying offset correction decreased the apparent range of pH variability by an amount that was mostly associated with instrumental drift, rather than physical and biogeochemical variability. Applying these offsets to the data decreased the range of pH measured by the ISFET sensor by approximately two thirds (Fig. 11), with new pHg standard deviations ranging between 0.009 and 0.048 within the top 150 m, and between 0.008 and 0.017 at greater depths.

- After applying this offset correction,  $pH_g$  was further corrected for *in situ* temperature and pressure using linear regression models. The method is outlined below:
  - 1. Calculate  $\Delta pH$  (Eq. (4)) as the difference between mean pHs and pHg.

$$\Delta p H = p H_{s,mean} - p H_g$$
(4)

- 2. Determine the line of best fit between  $\Delta pH$  and *in situ* temperature in the top 100 m of the water column where the temperature gradient was strongest using linear regression.
- 3. Correct  $pH_g$  for *in situ* temperature for the entire water column using the slope (*m*) and intercept (*c*) coefficients of the best fit line in step 2. to obtain  $pH_{g,tc}$ , where 'tc' stands for 'temperature corrected' values.
- 30 4. Calculate the difference between  $pH_{g,tc}$  profiles and mean  $pH_s$ , producing  $\Delta pH_{tc}$ , using an equation similar to Eq. (4).

- 5. Determine the line of best fit between  $\Delta p H_{tc}$  and pressure for the lower 900 m of the water column using linear regression.
- 6. Correct  $pH_{g,tc}$  for pressure for the entire water column using coefficients *m* and *c* in a similar way to step 3. to obtain  $pH_{g,tpc}$ , where 'tpc' stands for 'Temperature and Pressure corrected' values.

The derived equation used for correcting pHg is shown below:

5
$$pH_{g,tpc} = pH_g - 0.021 t/^{\circ}C + 4.5x10^{-5} P/dbar + 0.261$$
 (5)

where t is *in situ* temperature and P is pressure. A good fit was found between  $pH_g$  and *in situ* temperature, and a reasonable fit was found with pressure (Fig. 10). The standard deviations of  $pH_{g,tpc}$  ranged between 0.008 and 0.039 in the top 150 m of the water column and between 0.007 and 0.013 at greater depth, a further decrease in apparent variability of 13 to 23 % and 14 to 31 % respectively (Fig. 11).

- Johnson et al. (2016) ran a series of temperature and pressure cycling experiments when testing an ISFET pH sensor based on the Honeywell Durafet ISFET die. They found a temperature sensitivity of around  $\partial pH / \partial t = -0.018$ , similar to our calculated slope, and a pressure hysteresis of 0.5 mV (pH of around 0.01) at maximum compression (2000 dbar). This is equivalent to a pressure sensitivity of roughly  $\partial pH / \partial P = 5x10^{-6}$ , which is an order of magnitude smaller than in this study. This difference in pressure sensitivity could be related to the different housing materials used, as Johnson et al. (2016) used polyether ether
- 15 ketone (PEEK), whereas acrylic resin was used for our sensor.

Salinity covaries with temperature and pressure, and some of the salinity dependence of the offset between  $pH_s$  and  $pH_g$  might have been mis-attributed to the regression coefficients associated with temperature and pressure. The sensor characteristics should therefore be studied in detail under controlled laboratory conditions. However, for the purposes of calibrating the high-resolution, but poor-accuracy measurements (relative to  $pH_s$ ) obtained from the ISFET pH sensor, the present empirical

20 correction based on temperature and pressure appears to be sufficient to achieve a match to within the pH repeatability of the discrete samples of between 0.004 and 0.017.

**3.2.3 Coast to open ocean high resolution hydrographic and biogeochemical variability**

Spatial and temporal variability can be seen in pHg,tpc for three individual east-west transects using measurements obtained within different time periods (Fig. 12a-c). This pH variability is likely related to air-sea exchange of carbon dioxide (weak), changes in temperature (indirectly), and biological activity (Yao et al., 2016). In the top 100 m, pH higher than 8.12 was found at depths ranging from 20 to 95 m, whereas lower pH ranging from 8.06 to 8.09 were present closer to the surface at some locations (e.g. between 7.5 and 7.7° E, and east of 8° E). pH maxima were found at depths between 40 and 70 m, where  $\theta$  was around 15 °C (Fig. 12d-f), within the pycnocline (Fig. 12j-l). This band of high pH situated at 20 to 95 m depth corresponded with a thick layer of  $c(O_2)$  rich water at similar depths (Fig. 12m-o). pH and  $c(O_2)$  in the top 200 m of the water column more or

30 less followed isopycnal surfaces at a range of points in time and space. For example, the slanted isopycnals closer to the coast (east of  $7.95^{\circ}$  E), associated with geostrophic shear, corresponded with horizontal gradients in pH and  $c(O_2)$ . Below 100 m,

 $c(O_2)$  decreased to a minimum of < 170 µmol kg-1, which, although not spatially homogeneous, corresponded with generally colder, saltier, lower pH waters.

All three east-west transects can be separated into two parts roughly either side of 7.7° E for depths greater than 100 m. Lower pH of between 8.05 and 8.1 was found in the western part, whereas higher pH ranging from 8.07 to 8.12 was found in 5 the eastern part, which was partially seen in the pHs measurements (Fig. 5). The spatial variability of these east and west parts differed for each of the three time periods (times labelled in Fig. 12), with both the eastern high and western low pH patches changing in size vertically and horizontally, corresponding to spatial changes in  $\theta$  and salinity. Furthermore, salinity,  $\theta$ , and  $c(O_2)$  were lower in the western part, compared with values found at similar depths in the eastern section (Fig. 12d-i, 12m-o).

- In the top 100 m of the water column, variability in pH and  $c(O_2)$  are likely related to biological activity and air-sea gas 10 exchange. As discussed in Sect. 3.1, a DCM within this depth range is common in the Mediterranean Sea when waters are thermally stratified, and sufficient nutrients and light are available below the mixed layer (Estrada, 1996). High chlorophyll fluorescence was observed by the ship's sensor here (Fig. 4d). Enhanced  $c(O_2)$  at these depths are likely the by-product of photosynthesis, and the higher pH were likely the result of changes in the carbon equilibrium due to the consumption of CO2 (Cornwall et al., 2013; Rivaro et al., 2010; Copin-Montégut and Bégovic, 2002). A similar relationship between pH and
- 15 primary production was described by Álvarez et al. (2014) in the western Mediterranean Sea. As discussed in Sect 3.1, the fresher waters found in the top 100 m are likely MAW.

The difference in pH between the eastern and western parts at depths greater than 100 m depth highlighted the variability of water masses in this region. In particular, the higher pH found in the eastern part of the transect (east of 7.7° E), coinciding with high  $A_{\rm T}$  and c(DIC) (Fig. 5), was likely related to the flow of LIW, as described in Sect. 3.1. The LIW flows from the eastern

20 Mediterranean basin (east of the strait of Sicily), where pH is higher than in the western Mediterranean basin (Álvarez et al., 2014), towards the west along the continental shelf edge (Millot, 1999). This high pH found in the eastern section of the glider transect may therefore be remnants of these eastern Mediterranean waters. The low pH, low  $c(O_2)$  waters found deeper than 100 m results from increased respiration and remineralisation of organic matter (Lefèvre and Merlivat, 2012), coinciding with higher levels of c(DIC) deeper than 200 m (Merlivat et al., 2015), and which may have been more prominent in the western 25 part of the transect (west of 7.7° E) leading to decreased levels of pH.

The pycnocline shallowed east of 7.7° E in the top 100 m of the water column during all three time periods (times labelled in Fig. 12), which corresponded with shoaling high salinity, low pH, low  $c(O_2)$  waters, and high c(DIC),  $A_T$ , and chlorophyll fluorescence obtained by ship (Fig. 4 and Fig. 5). These features may be related to upwelling. Meteorological buoy M1 located south of the glider transect recorded an average surface wind direction of 198° towards the south-southwest which would be

- 30 favourable for coastal upwelling. However, the mean wind speed was only 2 m s-1 which is weak. On the other hand, salinity maxima seen at depths of 200 to 700 m seem to suggest an intrusion of LIW westward. An intrusion of water away from the coast towards the open ocean has been shown to increase divergence in regions close to shore with strong alongshore currents (Roughan et al., 2005). Upwelling signatures at this longitudinal range along the Sardinian coast have been simulated, particularly in the summer, by Olita et al. (2013) using a hydrodynamic 3D mesoscale resolving numerical model. They suggest
- 35 that a mixture of both current flow and wind preconditioned and enhanced upwelling in this region, which may have also been
the case during our deployment. Furthermore, chlorophyll fluorescence (Fig. 4) obtained by ship was higher closer to shore, indicative of a greater abundance of biomass in the top 100 m perhaps fuelled by upwelled nutrients (Porter et al., 2016; El Sayed et al., 1994).

**4 Conclusions**

10

15

20

- 5 Our trials of an experimental pH sensor in the Mediterranean Sea uncovered instrumental problems that were unexpected and will need to be addressed in future usage. These are summarised here:
  - 1. The data retrieved from the dual  $pH-p(CO_2)$  integrated sensor, and from the  $p(CO_2)$  unit of the stand-alone dual sensor could not be used due to quality issues. It is unclear why there was a problem with measurements obtained by the stand-alone  $p(CO_2)$  unit, however we think the regular on/off cycling of power to the integrated dual  $pH-p(CO_2)$  sensor in between sampling did not allow it to function properly. In future, we would suggest the addition of backup batteries to supply electricity to the sensor in between sampling.
  - 2. The stand-alone pH sensor was subject to drift. This could be reduced by subtracting a depth-constant time-varying offset from each dive using the difference between  $pH_g$  and  $pH_s$  at a more dynamically stable depth, but such an approach is not generally recommended or valid. We think that a change in E\* between the two n type silicon parts of the semiconductor might be the cause of the drift. To elucidate this drift further, in future two ISFET sensors should be tested in laboratory conditions within a bridge circuit to attempt to isolate possible factors contributing to drift. Focussing on the root cause of the sensor drift, rather than correcting the pH data for drift after the deployment, would be more beneficial to the longterm study of ISFET pH- $p(CO_2)$  sensors.
  - 3. The sensor was apparently affected by temperature and pressure, but it is unclear to what extent the empirical relationship between *in situ* temperature and  $\Delta pH$  in the thermocline (top 100 m) and between pressure and  $\Delta pH_{tc}$  in the deeper water (100 900 m) can be generalised.
  - 4. The effect of light caused the sensor to measure lower levels of  $pH_g$  in surface waters. This effect is expected to be ubiquitous wherever the sensor nears the surface during daytime. In future, the sensor will have to be positioned on the underside of the glider or equipped with a light shield to limit the effect of the sun when close to the surface.
- Despite the overall disappointing performance, we were able to demonstrate the potential use of the corrected glider pH measurements for uncovering biogeochemical variability associated with biological and physical mesoscale features.  $pH_g$ corrected for drift, temperature, and pressure, was compared temporally and spatially with other physical and biogeochemical parameters obtained by the glider. This comparison indicated that pH in the top 100 m of the water column was mostly related to biological activity, where  $c(O_2)$  was high. Below 100 m, low pH west of of 7.7° E was likely linked to the remineralisation of organic matter, whilst east of this point, higher pH may have been transported from the eastern Mediterranean basin via LIW.
  - 13

Shoaling isopycnals east of 7.7° E closer to shore may have been indicative of upwelling, and possible upwelling signatures at the same location could be seen in salinity,  $\theta$ , pH,  $c(O_2)$ , c(DIC),  $A_T$ , and chlorophyll fluorescence.

5

Acknowledgements. The authors would like to thank all partners who helped make REP14 - MED a success, the engineers, technicians and scientists onboard the NRV Alliance, NRV Planet, and those on land responsible for the logistics of the experiment, and the UEA glider science team for piloting the glider. We thank Bastien Queste and Gillian Damerell for help and support regarding the analysis of glider data. Michael Hemming's PhD project is funded by the Defence Science and Technology Laboratory (DSTL, UK) in close co-operation with Direction Générale de l'Armement (DGA, France), with oversight provided by Tim Clarke and Carole Nahum. We thank the Natural Environment Research Council (NERC, UK) for providing financial support for the demonstration of glider capability.